# Development of a dose-response model for porcine cysticercosis

**Daniel A. Andrade-Mogrovejo**[1☯], **Eloy Gonzales-Gustavson**[1,2☯]*, **Ana C. Ho-Palma**[3], **Joaquín M. Prada**[4], **Gabrielle Bonnet**[5], **Francesco Pizzitutti**[5], **Luis A. Gomez-Puerta**[1], **Gianfranco Arroyo**[6], **Seth E. O'Neal**[5,6], **Hector H. Garcia**[6,7], **Javier Guitian**[8], **Armando Gonzalez**[1], **Cysticercosis Working Group in Peru**[¶]

**1** Department of Animal and Public Health, School of Veterinary Medicine, Universidad Nacional Mayor de San Marcos, Lima, Peru, **2** Tropical and Highlands Veterinary Research Institute, Universidad Nacional Mayor de San Marcos, Lima, Peru, **3** Department of Human Medicine, School of Human Medicine, Universidad Nacional del Centro del Perú, Huancayo, Peru, **4** Department of Veterinary Epidemiology and Public Health, Faculty of Health and Medical Sciences, University of Surrey, Guildford, United Kingdom, **5** School of Public Health, Oregon Health & Science University and Portland State University, Portland, Oregon, United States of America, **6** Center for Global Health Tumbes, Universidad Peruana Cayetano Heredia, San Martín de Porres, Peru, **7** Cysticercosis Unit, National Institute of Neurological Sciences, Lima, Peru, **8** Veterinary Epidemiology, Economics and Public Health Group, Department of Pathobiology and Population Sciences, The Royal Veterinary College, Hertfordshire, United Kingdom

☯ These authors contributed equally to this work.
¶ Membership of the Cysticercosis Working Group in Peru is listed in the Acknowledgments.
* egonzalesg@unmsm.edu.pe

**Data Availability Statement:** All relevant data are within the paper and its Supporting information files. Besides, the R scripts to reproduce the results

## Abstract

*Taenia solium* is an important cause of acquired epilepsy worldwide and remains endemic in Asia, Africa, and Latin America. Transmission of this parasite is still poorly understood despite the design of infection experiments to improve our knowledge of the disease, with estimates for critical epidemiological parameters, such as the probability of human-to-pig infection after exposure to eggs, still lacking. In this paper, a systematic review was carried out and eight pig infection experiments were analyzed to describe the probability of developing cysts. These experiments included different pathways of inoculation: with ingestion of proglottids, eggs, and beetles that ingested eggs, and direct injection of activated oncospheres into the carotid artery. In these experiments, different infective doses were used, and the numbers of viable and degenerated cysts in the body and brain of each pig were registered. Five alternative dose-response models (exponential, logistic, log-logistic, and exact and approximate beta-Poisson) were assessed for their accuracy in describing the observed probabilities of cyst development as a function of the inoculation dose. Dose-response models were developed separately for the presence of three types of cysts (any, viable only, and cysts in the brain) and considered for each of the four inoculation methods ("Proglottids", "Eggs", "Beetles" and "Carotid"). The exact beta-Poisson model best fit the data for the three types of cysts and all relevant exposure pathways. However, observations for some exposure pathways were too scarce to reliably define a dose-response curve with any model. A wide enough range of doses and sufficient sample sizes was only found for the "Eggs" pathway and a merged "Oral" pathway combining the "Proglottids", "Eggs" and "Beetles" pathways. Estimated parameter values from this model suggest that a low infective dose is sufficient to result in a 50% probability for the development of any cyst or for

of the dose-response analysis can be found in the following repository in GitHub: https://github.com/dandradem/cysticercosis–dose-response.

**Funding:** AG and JG are funded by Concytec/FONDECYT (Reference 0247-2019-FONDECYT) and Medical Research Council (Reference MR/S025049/1) through a Newton-Paulet Fund project. Program URL: https://fondecyt.gob.pe/convocatorias/investigacion-cientifica/circulos-de-investigacion-en-salud-2018-01. This research was supported by the National Institute of Allergy and Infectious Diseases (R01 AI141554). HHG and GF are funded by the Fogarty International Center - US National Institutes of Health (grant number D43TW001140). Both institutions are within the US National Institutes of Health, URL: https://www.nih.gov/. The funders had no role in study design, data collection and analysis, decision to publish, or preparation of the manuscript.

**Competing interests:** The authors have declared that no competing interests exist.

viable cyst infections. Although this is a preliminary model reliant on a limited dataset, the parameters described in this manuscript should contribute to the design of future experimental infections related to *T. solium* transmission, as well as the parameterization of simulation models of transmission aimed at informing control.

## Introduction

Neurocysticercosis (NCC) is a common parasitic disease affecting the human central nervous system (CNS) and a leading cause of acquired epilepsy in endemic areas [1, 2]. The disease is caused by the larval stage of *Taenia solium* and is endemic in Latin American, African and Asian [3] countries where the presence of common risk factors, such as free-roaming pigs and poor sanitation, leads to high levels of disease transmission [1, 4]. The Foodborne Disease Burden Epidemiology Reference Group (FERG) under the World Health Organization (WHO) estimated that approximately 2.8 million disability-adjusted life years (DALYs) were lost globally in 2010 due to NCC [5] with heavy economic consequences such as an annual median loss of US $ 185.14 million in India, as estimated in 2011 [6]. *T. solium* also causes economic losses to the pig industry through decreased market value of infected pigs [2, 7–9].

The life cycle of *T. solium* involves humans as the only definitive host of the intestinal adult tapeworm and pigs as the intermediate host of the larval form or cysticercus [10]. Humans develop taeniasis by ingestion of undercooked pork infected with cysticerci [11], which then develop into adult tapeworms in the small intestine, releasing eggs or gravid proglottids in feces [12, 13]. In areas with inadequate sanitation and disposal of human feces, pigs ingest *T. solium* eggs/proglottids in human feces or in the surrounding contaminated environment and develop cysticerci mainly in muscles and subcutaneous tissue [2, 14, 15]. Humans can also serve as intermediate hosts through accidental ingestion of *T. solium* eggs released by a human tapeworm carrier [16]. The cysticerci can lodge anywhere in the human body leading to human cysticercosis [17]. When cysticerci establish in the CNS, the disease is called NCC and can manifest with seizures and other neurological symptoms [18].

The *T. solium* taeniasis/cysticercosis complex was determined to be eradicable by the Task Force for Disease Eradication (in 1992) [19] and was included under the WHO's "Global Plan to Combat Neglected Tropical Diseases (2008–2015)" [20]. The 2012 WHO roadmap for neglected tropical diseases established the following goals [21]: availability of a validated strategy for control and elimination of *T. solium* taeniasis/cysticercosis by 2015, and scaling up of interventions in selected countries by 2020. Since then, various control interventions have been developed in order to control or eliminate the disease [16, 22] and *T. solium* transmission models were also developed to evaluate the relative cost-effectiveness of these strategies [23–29]. However, development of these models is hampered by persistent limitations in the understanding of fundamental processes of transmission, such as the probability of human-to-pig infection upon exposure to eggs. In order to fill this gap, we carried out a systematic review of experimental pig infections, considering both published and unpublished studies. This review was used to develop a dose-response model with the purpose of estimating the probability of infection by porcine cysticercosis in experimental settings [30–36]. Fitting the results of these studies to a dose-response model provides a basis for translating exposure to *T. solium* eggs into a probability of infection.

**Table 1. Exposure pathways used in published experimental cysticercosis infection studies.**

| Ref. | Exposure pathways | Description |
|---|---|---|
| [30, 31] | Proglottids | The animals were trained to eat banana and honey balls. The required amount of gravid proglottids was then placed in such a ball and given to the pigs. |
| [30, 32] | Eggs | The required number of eggs was transferred into a gelatin capsule. Pigs were sedated and the capsules were introduced via an endoesophageal tube with 50–250 ml of support liquid. |
| [33, 34] | Beetles | Beetles of the *Ammophorus rubripes* species were fed cattle feces containing eggs three days before pig infection. Pigs were trained to eat banana pieces over a 3-day period. The required number of beetles was then placed in a banana piece and given to the pigs. |
| [35] | Carotid | *T. solium* eggs were exposed for 10 minutes to 0.75% sodium hypochlorite at 4°C to provoke hatching and oncosphere release. The oncospheres were then activated using artificial intestinal fluid at 37°C for 1 hour. Finally, pigs were anesthetized and inoculated with the required number of activated oncospheres via catheterization of the common carotid artery using the Seldinger technique and an ultrasound device with a vascular probe. |

## Methods

A systematic literature search on experimental infections of pigs with *T. solium* cysticercosis was conducted using indexed literature. In order to have comparable data, this search focused on articles in which the experimental inoculation was carried out through any the following approaches: 1) direct ingestion of gravid proglottids ("Proglottid") [30, 31], 2) inoculation, via an endoesophageal tube, of eggs placed in a gelatin capsule ("Eggs") [30, 32, 36], 3) direct ingestion of dung beetles previously fed with eggs ("Beetles") [33, 34], and/or 4) inoculation of activated oncospheres (AO) via catheterization of the common carotid artery ("Carotid") [35]. These procedures are described in more detail in Table 1. Selection was restricted to these pathways of inoculation because a systemic infection results from a single infective dose expressed in a known quantity of *T. solium* eggs or another unit of easy standardization.

Regarding the characteristics of experimentally infected pigs, only studies in which the pigs involved had received no treatment before and after inoculation were included. Furthermore, the pigs had to be between 1 and 2 months old, purchased from cysticercosis-free farms, and previously confirmed as negative for *T. solium* antibodies by enzyme-linked immunoelectro transfer blot (EITB) assay. After a sufficient period allowing for cysts to grow and become identifiable through macroscopic methods, all pigs must have been humanely euthanized and a detailed necropsy of the whole carcass implemented to determine the number of cysts they harbored (total, viable, and brain cysts). Language restrictions were applied, excluding articles written in a language other than those spoken or understood by the authors of this systematic review. The included languages were English, Spanish, French and Italian.

The selected database for this study was PubMed (https://pubmed.ncbi.nlm.nih.gov/) and the search was performed from December 1st 2020 until December 31st 2020.

### Search

The following search strategy was applied: In PubMed, using the Boolean operators "AND" and "OR", the terms "porcine cysticercosis", "porcine neurocysticercosis", "*Taenia solium*", "experimental infection" and "infection model" were introduced in the main search bar as (((porcine cysticercosis) OR (porcine neurocysticercosis)) AND (taenia solium)) AND ((experimental infection) OR (infection model)). Additionally, thesis research and

unpublished data from the Cysticercosis Working Group in Peru that were not found with this search strategy but matched the selection criteria after manual search were also included.

## Study selection

The articles were selected in two phases. The first phase consisted in the exclusion of articles based on a review of the title and/or abstract using the following exclusion criteria: 1) Studies that have not performed experimental infections, 2) Studies that performed experimental infections in other animal species, 3) Studies that performed experimental infections with other parasite species. In the second phase, full texts were read and studies selected according to the following selection criteria: 1) Availability of a detailed record of the cyst count (total, viable, and brain cysts), 2) Studies which performed experimental infections through any of the four exposure pathways previously mentioned, 3) Studies which performed a detailed necropsy of the whole animal carcass, 4) Studies with pigs meeting the criteria previously mentioned.

## Data collection

The data found in each selected paper were introduced in a database containing the following information: Author(s), year of publication, exposure pathway, identification of the animal, dose of the inoculum, number of viable cysts found in the carcass, number of degenerated cysts found in the carcass, total number of cysts found in the carcass, total number of cysts found in the brain.

## Standardization of dose units

In order to compare the different exposure pathways, all dose units were standardized to be expressed in eggs. In the studies that used proglottids, the infective dose for each proglottid was estimated by sampling with replacement a randomly-generated database composed of 10,000 observations following a discrete uniform distribution between 30,000–50,000 eggs for a full proglottid [13], and then linearly scaled for ½ and ¼ of a proglottid. For the "Beetles" pathway, we used the database composed by experimentally infected beetles with *T. solium* eggs from Gomez-Puerta *et al*. [37] to estimate the dose for each beetle by sampling with replacement. The database comprises 35 beetles, with a median of 21 eggs, a range of 3 to 235 eggs per beetle, and a right-skewed distribution (negative binomial distribution). Once the doses were expressed in *T. solium* eggs, the mean and standard deviation were calculated for each unique dose group for both the "Proglottids" and "Beetles" pathways. Studies using eggs placed in a gelatin capsule and carotid injection of oncospheres were assumed to be expressed in eggs. The standardized dose units for each exposure pathway are listed in S1 Table.

## Dose-response analysis

The dose-response analysis of the data was performed using R version 4.1.1 [38]. Five models of the way the probability of infection ($P_{inf}$) with any, viable or brain cysts varies as a function of the infective dose, in eggs (D), were evaluated for each exposure pathway. These include two-parameter log-logistic regression, logistic regression, the exponential model, and approximate and exact beta-Poisson models. These models were considered because their functions can accept a large range of doses (D) and are bound between 0 and 1, yielding a proportion (i.e., probability), which makes them ideal for binomial responses. Each dose-response model is briefly described below.

**Two-parameter log-logistic regression.** This model has two parameters: the slope of the curve ($\beta_{slope}$) and the median infective dose ($\beta_{ID50}$) (see Eq 1), which were estimated by maximum likelihood estimation (MLE) using the package 'drc' developed by Ritz and Streibig [39].

$$P_{inf}(D) = \frac{1}{1 + exp\left\{\beta_{slope}[log_e(D) - log_e(\beta_{ID50})]\right\}} \tag{1}$$

**Logistic regression.** This model has two parameters: the intercept ($\beta_0$) and the slope of the curve ($\beta_{slope}$) (see Eq 2). These parameters were estimated by MLE using the package 'stats' developed by the R Core Team [38, 40, 41].

$$P_{inf}(D) = \frac{1}{1 + exp\left[-\left(\beta_0 + \beta_{slope} * D\right)\right]} \tag{2}$$

**Exponential model.** This model has one parameter: the probability of at least one pathogen surviving the chain of barriers and causing an infection from any quantity of ingested pathogens (r) [42, 43] (see Eq 3). The parameter was estimated by non-linear least squares approximation using the package 'stats' developed by the R Core Team [38].

$$P_{inf}(D) = 1 - exp(-r * D) \tag{3}$$

This model, as well as the beta-Poisson models described below, makes the following three assumptions [44]: only one viable organism is required to produce the infection process, the exact number of organisms inoculated in each dose follows a Poisson distribution, and the survival of any organism in a single host is independent from the survival of other organisms in that host.

**Exact beta-Poisson.** Beta-Poisson models are a generalization of the exponential model and use the same three assumptions [44] described in the prior paragraph. Additionally, the exact beta-Poisson model incorporates heterogeneity in host-pathogen interactions by allowing for the probability of surviving the chain of barriers to infection to follow a Beta distribution of parameters alpha ($\alpha$) and beta ($\beta$) with the Kummer confluent hypergeometric function defined as $_1F_1(.,.,.)$ seen in Eq 4 [42, 43, 45]. The parameters were estimated following Xie *et al.*'s approach [45], which consisted in estimating the parameters by MLE and using them as prior information to generate, by a bootstrap algorithm of 5000 iterations, a sample-size-dependent confidence interval for each parameter in the model.

$$P_{inf}(D) = 1 - {_1F_1}(\alpha, \alpha + \beta, -D) \tag{4}$$

**Approximate beta-Poisson.** This model was derived using the approach in Furumoto and Mickey [46]. This approximation is used due to the mathematical complexity and difficulty in parameter estimation of the exact beta-Poisson model by transforming parameter beta ($\beta$) into a scale parameter (see Eq 5) [42, 45]. However, this approximation is only valid when two premises are fulfilled: $\beta \gg 1$, and $\alpha \ll \beta$ [42]. The parameters of this model, as for the exact beta-Poisson, were estimated following Xie *et al.*'s work [45].

$$P_{inf}(D) = 1 - \left(1 + \frac{D}{\beta}\right)^{-\alpha} \tag{5}$$

Whether the different types of dose-response models could be used to produce a dose-

response curve fitting relevant exposure pathways was first assessed by visual checking (for all models) then by estimating the value of model parameters. While this was done for all models, more attention was paid to estimating the parameters for models that, visually, seemed to have a chance of providing a good fit for the data. Subsequently, each dose-response curve was analyzed through the calculation of both the residual sum of squared errors of prediction (SSE) and the coefficient of determination ($R^2$) to determine the type of model that best fits the data. For both beta-Poisson models, only the median probability of infection resulting in the posterior distribution was considered and compared to observed data to develop these calculations. Using the best-fitted model, we estimated the doses required to produce a probability of infection of 1% (ID01) and 50% (ID50) for the development of any, viable only, and brain cysts according to the exposure pathways.

Finally, the data from all three oral exposure pathways (i.e., "Proglottids", "Eggs" and "Beetles") were analyzed together to build a single "Oral" pathway with the purpose of obtaining a more robust model with a wider range of observations reflecting the plausible natural infection pathways of the disease. The resulting "Oral" pathway was analyzed using the best-fitted type of dose-response model.

## Results

### Study selection

The review process and the number of articles selected at each stage of the review is shown in Fig 1. From an initial number of 166 articles, only 8 studies that performed experimental

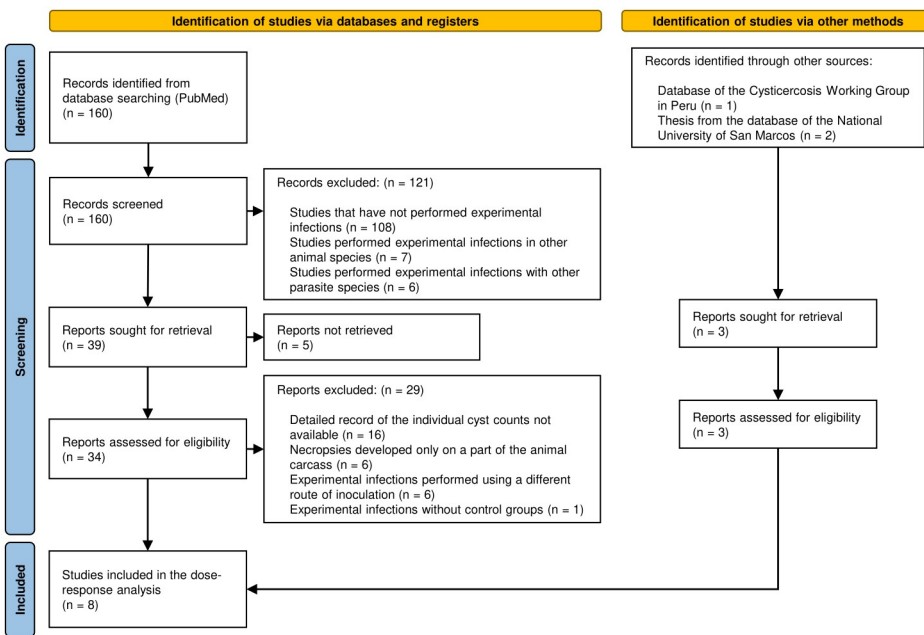

**Fig 1. PRISMA 2020 flow diagram for new systematic reviews which included searches of databases, registers and other sources.** *Consider, if feasible to do so, reporting the number of records identified from each database or register searched (rather than the total number across all databases/registers). **If automation tools were used, indicate how many records were excluded by a human and how many were excluded by automation tools. *From*: Page MJ, McKenzie JE, Bossuyt PM, Boutron I, Hoffmann TC, Mulrow CD, et al. The PRISMA 2020 statement: an updated guideline for reporting systematic reviews. BMJ 2021;372:n71. doi: 10.1136/bmj.n71. For more information, visit: http://www.prisma-statement.org/.

**Table 2. Cysticercosis challenge data by exposure pathway.**

| Exposure pathway | Inoculum dose | Exposed pigs | Infected pigs with any cyst | Infected pigs with viable cysts | Infected pigs with brain cysts |
|---|---|---|---|---|---|
| Proglottids | 0 proglottids | 3 | 0 (0%) | 0 (0%) | 0 (0%) |
| | ¼ proglottid | 8 | 8 (100%) | 7 (87.5%) | 1 (12.5%) |
| | ½ proglottid | 8 | 7 (87.5%) | 5 (62.5%) | 3 (37.5%) |
| | 1 proglottid | 30 (14)[a] | 28 (93.3%) | 27 (90%) | 7 (50%) |
| Eggs | 0 eggs | 7 | 0 (0%) | 0 (0%) | 0 (0%) |
| | 10 eggs | 4 | 3 (75%) | 1 (25%) | 0 (0%) |
| | 100 eggs | 9 | 6 (66.7%) | 2 (22.2%) | 0 (0%) |
| | 1000 eggs | 11 | 10 (90.9%) | 3 (27.3%) | 0 (0%) |
| | 10000 eggs | 9 | 9 (100%) | 9 (100%) | 2 (22.2%) |
| | 20000 eggs | 5 | 5 (100%) | 5 (100%) | 4 (80%) |
| | 100000 eggs | 5 | 5 (100%) | 5 (100%) | 3 (60%) |
| Beetles | 1 beetle | 6 | 5 (83.3%) | 4 (66.7%) | 1 (16.7%) |
| | 3 beetles | 6 | 6 (100%) | 6 (100%) | 1 (16.7%) |
| | 4 beetles | 8 | 6 (75%) | 2 (25%) | 0 (0%) |
| | 6 beetles | 30 | 24 (80%) | 23 (76.7%) | 2 (6.7%) |
| Carotid | 2500 AO | 5 | 5 (100%) | 5 (100%) | 3 (60%) |
| | 5000 AO | 6 | 6 (100%) | 6 (100%) | 6 (100%) |
| | 10000 AO | 11 | 9 (81.8%) | 9 (81.8%) | 7 (63.6%) |
| | 45000 AO | 1 | 1 (100%) | 1 (100%) | 1 (100%) |
| | 50000 AO | 5 | 5 (100%) | 5 (100%) | 4 (80%) |

Proglottids: direct ingestion of gravid proglottids; Eggs: inoculation via an endoesophageal tube of eggs placed in a gelatin capsule; Beetles: direct ingestion of beetles previously fed with eggs; Carotid: inoculation of activated oncospheres via catheterization of the common carotid artery; AO: activated oncospheres.

[a] Brain cyst counts were only available for 14 pigs.

infections were included in this review (including the unpublished study): 2 studies using the "Proglottid" pathway [30, 31], 4 studies using the "Eggs" pathway [30, 32, 36], 3 studies using the "Beetles" pathway [33, 34], and 2 studies using the "Carotid" pathway [35].

After combining the data from all studies, the combined sample includes a total of 177 pigs divided in 20 unique dose groups (including control groups). The dose groups are listed in Table 2 with information on the exposure pathway used for inoculation, number of exposed pigs, and number of infected pigs with any cyst (at least one viable and/or degenerated cyst in the whole carcass), with viable cysts (at least one viable cyst in the whole carcass) and with brain cysts (at least one viable and/or degenerated cyst in the brain). A table listing the dose groups with their respective reference (both published and unpublished studies) can be found in S2 Table.

All studies obtained gravid *T. solium* proglottids from taeniasis patients after treatment with 2g of niclosamide, most preserved them in a saline solution with antibiotics at a temperature between 4–5°C until needed for infection [30–36], except for the study by Gomez-Puerta et al. [33] which preserved the gravid proglottids in 25% glycerol supplemented with antibiotics. Additionally, the collection of *T. solium* eggs from the proglottids were performed by gentle homogenization [30, 35], maceration [32, 36] and direct mixing with cattle feces [33, 34]. A more detailed description of the experimental designs is available in the publications referenced in Table 1.

The unpublished study conducted by the Cysticercosis Working Group in Peru included in this paper was reviewed and approved by the Institutional Ethics Committee for the Use of

Animals (Comite de Ética y Bienestar Animal (CEBA)) at the School of Veterinary Medicine, Universidad Nacional Mayor de San Marcos (protocol no. 2009–003, 2010–012, and 2012–004), and by the Institutional Ethics Committee for the Use of Animals (Comité Institucional de Ética para el Uso de Animales (CIEA)) at the Universidad Peruana Cayetano Heredia (protocol no. A5146-01), both located in Lima, Peru.

Even before proceeding to a formal dose-response analysis, we see that there is a lack of a visible trend in the way the share of infected pigs evolves with the doses available for the "Beetles" and "Carotid" pathways, as well as for the "Proglottids" pathway except possibly for the development of brain cysts. The low size of available pig samples and limited range in doses likely explain these results. Hence, we focused our analysis on the "Eggs" pathway as well as on the combined "Oral" pathway (eggs + proglottids + beetles), which groups information from the three pathways that most closely mimic wild-type infection routes. The combined pathway uses more data points hence it is hoped it can allow for more robust estimations.

## Dose-response analysis

The R scripts to reproduce the results of the dose-response analysis can be found in the following repository in GitHub: https://github.com/dandradem/cysticercosis--dose-response. Among all models assessed, the exact beta-Poisson was found to fit the data with the greatest precision using the least squares method and the coefficient of determination ($R^2$) (see S1 Appendix), and its parameters are presented in this section while the results of the other types of dose-response models can be found in S2–S5 Appendices. Further, as explained earlier, the number of data points available in the literature has often proven insufficient to develop a model, and only the "Eggs" and the "Oral" pathways allowed for the development of dose-response models. The MLEs for the parameters of these curves are presented in Table 3, while the results for the other exposure pathways are shown in S6 Appendix. Moreover, with the approximate beta-Poisson model, we found parameters similar to those of the exact beta-Poisson model, but they did not fulfill the premises previously mentioned ($\beta \gg 1$ and $\alpha \ll \beta$) in most of the dose-response curves and presented misleading results at the beginning of the curve as shown in S5 Appendix.

For the development of any type of cyst through these exposure pathways, the doses required to achieve 50% and 80% probabilities of infection are low: lower than 10 and 100 *T. solium* eggs respectively (see Table 4). Higher doses are necessary to obtain both viable and brain cysts using these pathways (Figs 2 and 3). This pattern is confirmed by the doses required

**Table 3. MLEs for the parameters of the exact beta-Poisson model.**

| Exposure pathway | α | β |
|---|---|---|
| **Development of any (viable or degenerated) cyst** | | |
| Oral | 0.227 | 0.286 |
| Eggs | 0.371 | 1.41 |
| **Development of viable cysts** | | |
| Oral | 0.213 | 2.70 |
| Eggs | 0.792 | 500.00 |
| **Development of brain cysts** | | |
| Oral | 0.119 | 303.15 |
| Eggs | 0.136 | 500.00 |

α: parameter alpha; β: parameter beta; Oral: oral inoculation; Eggs: inoculation via an endoesophageal tube of eggs placed in a gelatin capsule.

**Table 4. Estimated doses necessary to cause a 1% and 50% probability (median and 95% range) for the development of any, viable and brain cysts, by exposure pathway.**

| Exposure pathway | ID01 | ID50 |
|---|---|---|
| **Dose of eggs required for the development of any (viable or degenerated) cyst** | | |
| Oral | 0.024 (0.012–0.123) | 2.78 (0.954–19.63) |
| Eggs | 0.056 (0.011–0.497) | 8.49 (0.830–49.77) |
| **Dose of eggs required for the development of viable cysts** | | |
| Oral | 0.149 (0.027–0.658) | 59.94 (5.85–183.07) |
| Eggs | 7.39 (1.51–24.77) | 739.07 (253.53–1963.04) |
| **Dose of eggs required for the development of brain cysts** | | |
| Oral | 28.48 (6.42–166.81) | $9.33 \times 10^5$ ($1.59 \times 10^4$–$1.79 \times 10^7$) |
| Eggs | 43.29 (2.42–422.92) | $6.42 \times 10^4$ (8697.49–$4.75 \times 10^9$) |

ID01: estimated dose to cause produce a probability of infection of 1%; ID50: estimated dose to cause produce a probability of infection of 50%; Oral: oral inoculation; Eggs: inoculation via an endoesophageal tube of eggs placed in a gelatin capsule.

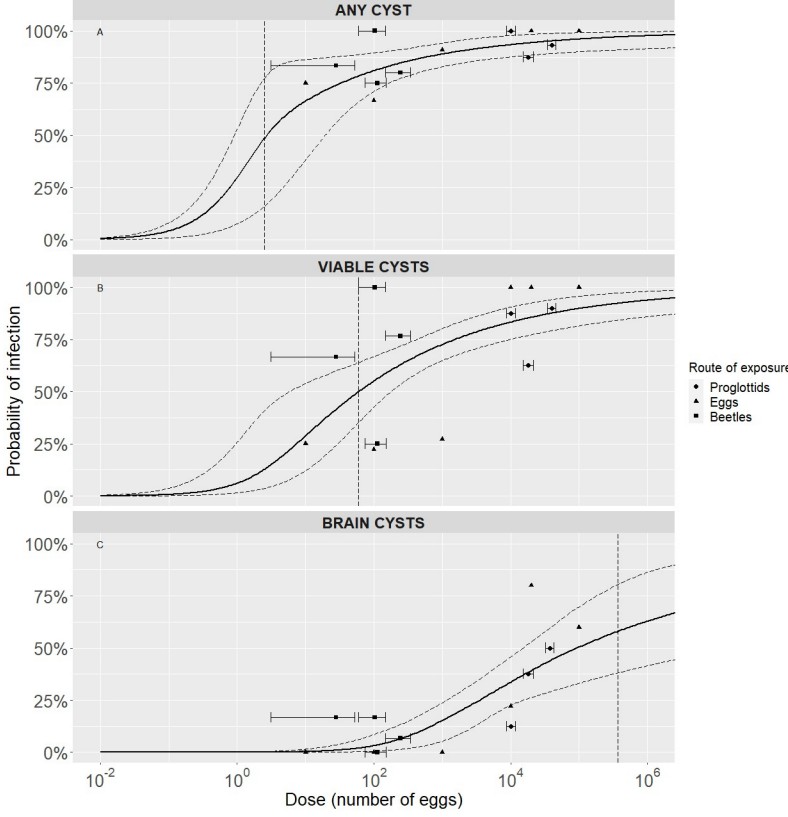

**Fig 2. Exact beta-Poisson dose-response relationship for the "Oral" pathway by type of cyst.** Each graph shows the median (solid black curve) and 95% range (dashed black curves) of the probability of infection as a function of dose, median ID50 infectious dose (dashed black vertical line), and the available data point with its standard deviation ("Proglottids" and "Beetles" pathways only). (A) Development of any (viable or degenerated) cyst. (B) Development of viable cysts. (C) Development of brain cysts.

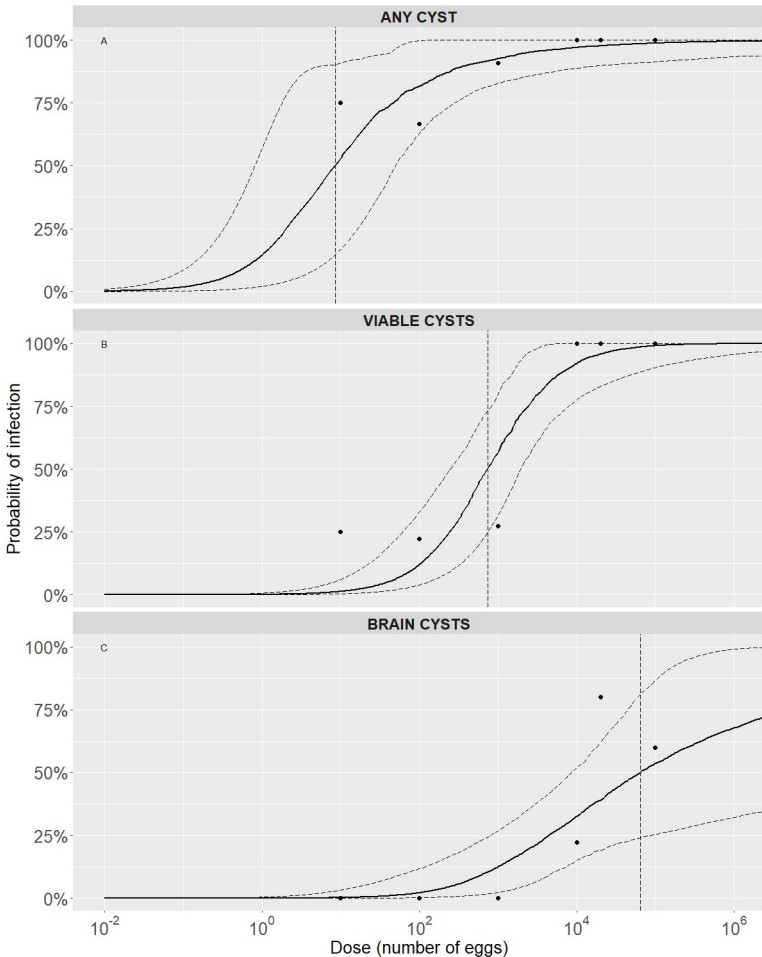

**Fig 3. Exact beta-Poisson dose-response relationship for the "Eggs" pathway by type of cyst.** Each graph shows the median (solid black curve) and 95% range (dashed black curves) of the probability of infection as a function of dose, median ID50 infectious dose (dashed black vertical line), and the available data point. (A) Development of any (viable or degenerated) cyst. (B) Development of viable cysts. (C) Development of brain cysts.

to cause a 1% and 50% probability of infection presented in Table 4. The development of viable cysts required doses above 50 eggs to achieve a 50% probability of infection, while the doses required to obtain brain cysts with the same probability were greater than 60000 eggs (Table 4). Finally, please note that some of the computations give infective doses below one egg. What a fractional dose of 0.056 eggs to reach ID01 means is that giving one egg to each pig in a sample would lead to an expected share of infected pigs higher than one percent. To infect 10 pigs out of a sample of 1000 (share of 1%), for example, one may need to give one egg to 56 pigs and none to the other 944 pigs.

## Discussion

We analyzed several dose-response models in order to estimate the probability of infection according to the dose of *T. solium* eggs given to pigs through different exposure pathways. Due to the limited range of doses used in the experimental infection collected through the systematic review, we could not estimate a dose-response model for all the exposure pathways. We

were successful only using the "Oral" pathway and the "Eggs" pathway. A dose-response relationship was best fitted for these exposure pathways using the exact beta-Poisson model. This model has a maximum risk curve which limits the upper confidence level through the inclusion of the Kummer confluent hypergeometric function, which also takes into account the heterogeneity of the host-pathogen interaction (Eq 4) [42, 43, 47], and thus is widely used for risk analysis to estimate the probability of infection by diseases caused by viruses, bacteria and other macro parasites [42]. This model also described the observed experimental data with better precision. On the other hand, the approximate beta-Poisson model has a scalable relation by its parameter β (Eq 5), and therefore lacks a maximum risk curve. As a consequence, it can produce misleading results when the premises for its validation are not fulfilled (S5 Appendix) [42].

To be comprehensive, results for the "Proglottids", "Beetles" and "Carotid" pathways are presented in S6 Appendix. However, because of the low number of studies using low doses, proper dose-response curves could not be developed for these pathways: a striking example is the "Carotid" pathway which leads to infections in all pigs in the sample for almost all doses available. Another issue regarding the development of dose response curves is when the probability of infection is below 100% at high doses, corresponding to an immunity plateau and an overestimation of the probability of infection at low doses (S6 Appendix) [48]. Consequently, the inclusion of an immune parameter in the exact beta-Poisson function is highly recommended for future dose-response analysis to obtain best-fitted models. As compared to the other pathways, the "Eggs" pathway was informed by a broader range of experimental data, including a number of experiments with small doses of eggs, with 10 eggs leading to successful infection in some but not all of the pigs. This resulted in a flattening of the slope of the curve at the lowest estimated doses as compared to what could have been obtained with other pathways. Besides, this pathway presented observed data with probabilities of infection for the development of both any and viable cysts equal to 100% at high doses. Therefore, we believe that the dose-response curve developed for the "Eggs" pathway can estimate the probability of infection at lower doses in a reasonably realistic way as compared to other pathways, even without the inclusion of additional parameters. We also note that, even as the data available do not allow for the development of dose-response curves for pathways other than "Eggs", data available suggests that curves would be displaced to the right when considering viable cysts, with the exception of the "Carotid" pathway, for which the probability of infection was extremely high for both all and viable cysts (S6 Appendix).

Two-parameter log-logistic regression, logistic regression and the exponential model were also assessed. They could not be fitted to the "Proglottids", "Beetles" and "Carotid" pathways, due to the limited information in the dataset and the magnitudes of the doses used in the studies. The exponential model had the best performance during the evaluation of these models overall (S2–S4 Appendices). With the purpose of improving the fit to the data, logistic regression was also evaluated assuming a negative control group, with a zero dose and zero probability of infection, composed of 15 animals (S3 Appendix). This assumption was useful because this model does not consider a probability of infection of 0% when an infectious dose is not administered [41]. However, the resulting model still did not fit the data appropriately, showing that it is not suitable to describe the probability of infection by porcine cysticercosis.

We chose to merge the "Eggs", "Beetles" and "Proglottids" pathways into a combined "Oral" pathway. The aim was to obtain a more robust dose-response relationship, as most of these pathways have limited dose ranges that are in distinct ranges, e.g., in the 10,000s for "Proglottids" and the hundreds for "Beetles", and between ten and 100,000 for "Eggs", making fitting a dose-response model accurately with only one of these pathways challenging. These three exposure pathways provide parameters that are similar. Further, they all involve the

digestive system, which the "Carotid" pathway bypasses. The "Oral" pathway is likely the closest to the wild-type exposure pathway for *T. solium* cysticercosis in pigs (ingestion of feces with proglottids and/or eggs, and possibly isolated eggs and/or insects) [2].

Most pigs in natural endemic conditions are infected with less than ten cysts [16], while few eggs are enough to infect many pigs in our dose-response model. These findings suggest that other pathways rather than ingestion of gravid proglottids may play a role in the dynamics of the disease and it would be useful to study other sources of eggs such as the soil, water, or vectors with eggs [15, 49]. Considering the possible importance of low doses of eggs in pig infection, novel methods with a limit of detection of 200 eggs per 200 g (1 egg per gram of sample (EPG)) validated by Gamboa *et al.* [50], and the Droplet Digital PCR with a limit of 10 eggs per 5 g (2 EPG) [51], would be recommended to evaluate *T. solium* eggs in the environment.

Several studies have demonstrated the capability of arthropods such as flies, dung beetles and blowflies, to transmit taeniid tapeworm eggs [33, 52, 53]. In the case of *T. solium*, the potential of dung beetles in the transmission of *T. solium* eggs has been evaluated in multiple studies [33, 34, 37]. However, these studies only used beetles previously fed with eggs *in vitro* and thus transmission in the environment with dung beetles carrying eggs in natural conditions has yet to be described. Furthermore, Gonzalez *et al.* has proposed another secondary transmission mechanism suggesting that pigs ingesting a gravid proglottid may spread the eggs in the environment through their feces, infecting other animals in the herd [54]. If this is true, spontaneous expulsion of the whole tapeworm by a carrier in the interval between diagnostic and treatment in control studies, which may be common [55], could be a major contributor (due to the number of eggs involved) to this contamination. In conjunction with the high infectivity of eggs shown in this study, these processes may help *T. solium* maintain its endemic stability and return to the original levels of disease after the completion of control programs if these are not focused on the treatment of both human and porcine population [54, 56]. Enhancing biosecurity levels to mitigate the risk of accidental ingestion of eggs is also recommended based on the high infectivity of *T. solium* eggs observed in this study.

Despite the presence of cysts in the pigs' brain represents a null impact on public health, the models for the development of brain cysts can also serve as reference to estimate the required doses to obtain cysts in pigs' brain experimentally for future studies with the aim of evaluating treatment schemes for NCC using pigs as animal models. On the other hand, the model for development of brain cysts through the "Oral" pathway may be extrapolated to human NCC based on recent evidence which suggests that oncospheres are distributed to all tissues in humans, instead of being established preferentially in the brain, as was believed before [18]. However, it is necessary to demonstrate that there is no difference in the distribution of cysts between humans and pigs in order to carry out such extrapolation.

The main limitations in this study were the limited number of data points and range of doses to develop a robust model. Further, the low sample sizes in some of the studies led to an increase of the uncertainty in the corresponding sections of the curve. It is necessary to carry out experimental infections that will allow us to evaluate the performance of infection at lower doses in controlled scenarios with less uncertainty in the quantity supplied and controlling the viability of the inoculum. These improvements may inform future models in order to reduce the uncertainty in the parameters described in this paper.

In conclusion, the exact beta-Poisson is proposed as a potential continuous dose-response model to estimate the probability of infection across a range of doses of multiple orders of magnitude, rather than limiting the assessment of human-to-pig transmission to narrow scenarios. Although data available is limited, the parameters of the model could be used to estimate the doses necessary to obtain infected pigs with the type of cysts required for the development of studies involving experimental infections. Moreover, the parameters could

also be considered in the development of future quantitative risk assessments to fill the gap in the probability of human-to-pig transmission [57]. Given the links between infection and immunity [58], knowing the dose required to produce an infection may also help estimate the doses required to produce protective immunity. Further, in future work, the current framework could be expanded to explore the correlation between the inoculated dose and the number of cysts in pigs' bodies and brains and even link the infectious dose to the cyst distribution in the pig, with the purpose of further describing the disease. The results obtained in this pig model might be extrapolated (to a certain extent) to human cysticercosis provided both hosts have a similar cyst distribution. Finally, this paper may provide useful insights when trying to develop new interventions and models. In this context, the parameters of the exact beta-Poisson for the development of viable cysts through oral inoculation may be the most useful, because this exposure pathway best represents the natural infection pathway, and viable cysts have the potential to produce human taeniasis perpetuating the *T. solium* life cycle [18, 35].

## Supporting information

**S1 Table. Standardized dose units for each exposure pathway.**
(DOCX)

**S2 Table. Cysticercosis challenge studies with references.**
(DOCX)

**S3 Table. Raw data used for the dose-response analysis.**
(CSV)

**S1 Appendix. Assessment of the fit of the dose-response models.**
(DOCX)

**S2 Appendix. Two-parameter log-logistic regression model.**
(DOCX)

**S3 Appendix. Logistic regression model.**
(DOCX)

**S4 Appendix. Exponential model.**
(DOCX)

**S5 Appendix. Approximate beta-Poisson model.**
(DOCX)

**S6 Appendix. Exact beta-Poisson model for "Proglottids", "Beetles", and "Carotid" pathways.**
(DOCX)

## Acknowledgments

We are grateful to PhD. Gertjan Medema from the KWR Water Research Institute for his valuable and helpful comments regarding the interpretation of the developed dose-response models.

## Members of the Cysticercosis Working Group in Peru

Robert H. Gilman, MD, DTMH; Manuela Verastegui, PhD; Mirko Zimic, PhD; Javier Bustos, MD, MPH; and Victor C. W. Tsang, PhD (Coordination Board); Silvia Rodriguez, MSc; Isidro

Gonzalez, MD; Herbert Saavedra, MD; Sofia Sanchez, MD, MSc, Manuel Martinez, MD (Instituto Nacional de Ciencias Neurologicas, Lima, Peru); Saul Santivanez, MD, PhD; Holger Mayta, PhD; Yesenia Castillo, MSc; Monica Pajuelo, PhD; Luz Toribio; Miguel Angel Orrego, MSc (Universidad Peruana Cayetano Heredia, Lima, Peru); Maria T. Lopez, DVM, PhD; Cesar M. Gavidia, DVM, PhD; Ana Vargas-Calla, DVM (School of Veterinary Medicine, Universidad Nacional Mayor de San Marcos, Lima, Peru); Luz M. Moyano, MD; Ricardo Gamboa, MSc; Claudio Muro; Percy Vichez, MSc (Cysticercosis Elimination Program, Tumbes, Perú); Sukwan Handali, MD; John Noh (Centers for Disease Control, Atlanta, GA); Theodore E. Nash, MD (NIAID, NIH, Bethesda, MD); Jon Friedland (Imperial College, London, United Kingdom).

## Author Contributions

**Conceptualization:** Daniel A. Andrade-Mogrovejo, Eloy Gonzales-Gustavson.

**Data curation:** Daniel A. Andrade-Mogrovejo, Eloy Gonzales-Gustavson, Ana C. Ho-Palma, Joaquín M. Prada.

**Formal analysis:** Daniel A. Andrade-Mogrovejo, Eloy Gonzales-Gustavson, Ana C. Ho-Palma, Joaquín M. Prada.

**Funding acquisition:** Seth E. O'Neal, Hector H. Garcia, Javier Guitian, Armando Gonzalez.

**Investigation:** Daniel A. Andrade-Mogrovejo, Eloy Gonzales-Gustavson, Ana C. Ho-Palma.

**Methodology:** Daniel A. Andrade-Mogrovejo, Eloy Gonzales-Gustavson, Ana C. Ho-Palma, Joaquín M. Prada, Gabrielle Bonnet, Francesco Pizzitutti, Luis A. Gomez-Puerta, Gianfranco Arroyo.

**Project administration:** Eloy Gonzales-Gustavson, Seth E. O'Neal, Hector H. Garcia, Javier Guitian, Armando Gonzalez.

**Resources:** Seth E. O'Neal, Hector H. Garcia, Javier Guitian, Armando Gonzalez.

**Supervision:** Daniel A. Andrade-Mogrovejo, Eloy Gonzales-Gustavson, Ana C. Ho-Palma, Joaquín M. Prada, Gabrielle Bonnet, Francesco Pizzitutti, Luis A. Gomez-Puerta, Gianfranco Arroyo, Seth E. O'Neal, Hector H. Garcia, Javier Guitian, Armando Gonzalez.

**Validation:** Daniel A. Andrade-Mogrovejo, Eloy Gonzales-Gustavson, Ana C. Ho-Palma, Joaquín M. Prada, Gabrielle Bonnet, Francesco Pizzitutti, Luis A. Gomez-Puerta, Gianfranco Arroyo, Seth E. O'Neal, Hector H. Garcia, Javier Guitian, Armando Gonzalez.

**Visualization:** Daniel A. Andrade-Mogrovejo, Gabrielle Bonnet, Francesco Pizzitutti, Luis A. Gomez-Puerta, Gianfranco Arroyo.

**Writing – original draft:** Daniel A. Andrade-Mogrovejo, Eloy Gonzales-Gustavson.

**Writing – review & editing:** Daniel A. Andrade-Mogrovejo, Eloy Gonzales-Gustavson, Ana C. Ho-Palma, Joaquín M. Prada, Gabrielle Bonnet, Francesco Pizzitutti, Luis A. Gomez-Puerta, Gianfranco Arroyo, Seth E. O'Neal, Hector H. Garcia, Javier Guitian, Armando Gonzalez.

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
