## [Decision Letter · Decision Letter 0]

9 Sep 2021

PONE-D-21-23111Development of a dose-response model for porcine cysticercosisPLOS ONE

Dear Dr. Gonzales-Gustavson,

Thank you for submitting your manuscript to PLOS ONE. After careful consideration, we feel that it has merit but does not fully meet PLOS ONE’s publication criteria as it currently stands. Therefore, we invite you to submit a revised version of the manuscript that addresses the points raised during the review process.

We look forward to receiving your revised manuscript.

Kind regards,

Brecht Devleesschauwer

Academic Editor

PLOS ONE

Journal Requirements:

https://journals.plos.org/plosone/s/file?id=ba62/PLOSOne_formatting_sample_title_authors_affiliations.pdf\\

**Additional Editor Comments:**

The manuscript has been assessed by three reviewers, including experts in Taenia solium and in dose-response modelling. All three experts provided detailed comments and suggestions to improve the manuscript. Reviewer #3 in particular raised a series of methodological and statistical concerns which would require substantial revisions.

If you decide to develop and submit a revised version of the manuscript, this will be reassessed by the initial reviewers.

In your revision note, please include EACH of the reviewer comments, provide your reply, and when relevant, include the modified/new text (or motivate why you decided not to modify the text). Note that failure to do so will result in a rejection of the manuscript.

Reviewers' comments:

Reviewer's Responses to Questions

**Comments to the Author**

1. Is the manuscript technically sound, and do the data support the conclusions?

Reviewer #1: Yes

Reviewer #2: Yes

Reviewer #3: No

2. Has the statistical analysis been performed appropriately and rigorously? 

Reviewer #1: Yes

Reviewer #2: Yes

Reviewer #3: No

3. Have the authors made all data underlying the findings in their manuscript fully available?

Reviewer #1: Yes

Reviewer #2: Yes

Reviewer #3: Yes

4. Is the manuscript presented in an intelligible fashion and written in standard English?

Reviewer #1: Yes

Reviewer #2: Yes

Reviewer #3: Yes

5. Review Comments to the Author

Reviewer #1: I would like to congratulate the authors on a very interesting and timely piece of research in the T.solium field. I do however have some substantial comments that I think should be addressed:

Intro

• Can the authors be more specific with the phrase “with heavy economic consequences” on line 99

• The authors suggest the route of exposure for pigs is via ingestion of egg/proglottids in stools, however exposure may occur through consumption of eggs that have been dispersed in the environment (not in stools) via mechanical vectors. This is also mentioned as a “route of exposure” in the experimental studies in the methods (Table 1), so providing some context in the intro would be useful. See

Jansen, F., Dorny, P., Gabriël, S. et al. The survival and dispersal of Taenia eggs in the environment: what are the implications for transmission? A systematic review. Parasites Vectors 14, 88 (2021). https://doi.org/10.1186/s13071-021-04589-6

Methods:

• Lines 128 onwards: It is not clear whether the search to identify cysticercosis challenge study data was conducted in a structured way (i.e., with a systematic search methodology) or non-structured? If structured, a flow diagram to identify the number of studies identified at each stage (i.e., title, abstract, full-text using the PRISMA method) should be included. If non-structured, I would suggest this literature search should be re-conducted using a systematic methodology, to highlight search terms/ literature databases so this search can be replicated (or updated in the future), and identify whether other selection criteria including language exclusion criteria etc to avoid reporting biases.

• Can the authors explicitly say whether the criteria for pig selection (lines130 – 139) were used for pig selection in the CWG unpublished study (line 153) – I get this is true from the first paragraph in the methods, but think a few words on line 153 to this effect would improve clarity.

• I do not understand why the number of exposed pigs varies so much between inoculum dose for each route of exposure in Table 2 CWG challenge study? Are there any sample size calculations behind the number of exposed pigs?

• Table S1 refers to the route-dose-exposure-infection numbers from studies collated (and combined) from the literature (i.e. not including CWG unpublished study)? It is not clear in the methods text (line 160).

• I do not think line 178-183 are necessary if this is a secondary analysis of published studies, but in the case of the CWG unpublished study there should be reference to ethical approval only.

• Was the distribution for eggs from beetles a log-normal distribution to reflect the skewedness (it would be useful for the authors to be more explicit here on lines 191-192)

• What was the criteria for determining that the five dose-response models could produce a dose-response curve fitting (was this just a visual check, or any other diagnostics used?) on lines 249-251?

• I do not see the probability of surviving the chain of barriers to infection (r) on lines 233-234 in equation 5, or perhaps I am misunderstanding this function?

• Are there any reasons for selecting minimum doses required to produce a probability of infection of 1% and 50% (and not including higher probabilities i.e. probability of infection of 95%?) on lines 253-255

• Can the authors provide further details on how the three oral routes of exposure were merged into the single “oral” pathway in lines 256-257?

Results:

• So I am clear, the beta-Poisson model was found to best fit the data for all exposure routes in lines 262-263

• Can the authors combine supplementary tables S3 and S4 comparing model fits with either SEE or R2 to facilitate an easier comparison?

• The axis numbers (especially on the x-axis) of figures 1 – 4 are very hard to see without zooming in substantially, please could these be increased in size.

• I am unclear how the statement “For the development of any type of cyst, the four routes of infection presented a high probability of infection at low doses (Table 4)” (lines 297 – 300) can be obtained from table 4, as this only looks 1 and 50% probabilities of infection (without referring to the figures?), and equally the next sentence seems to suggest this can be read from Table 4, but can only really be read from Figures 1-3 if I am understanding correctly, so I think to improve clarity it would be useful to show these sentences indicate when referring to the figures (throughout lines 297 – 309)?

• I do not understand where the “half the inoculated pigs” comes from in the “to obtain viable cysts in at least half of the inoculated pigs” (lines 304-303); is this what ID50 is measuring in terms of the probability of infection in at least 50% of the exposed pigs? (Although I thought this was the minimum dose to obtain a 50% probability of infection in any pig)?

• I am not sure what the phrase “notorious reduction” means on line 306

• I do not understand what the sentence “even though one of these routes of exposure bypasses the mouth (“Eggs”)” (line 322)” – I would have thought the eggs are only consumed by the pigs via the mouth (unless the authors are talking about the Carotid inoculation pathway?)

• However the ID50 infectious dose is >100 for any cyst in the oral pathway (4A) but <100 for any cyst in the carotid pathway, which is an important distinction re lines 337-339, so I am not sure the median probabilities are so comparable (although the overall shape is similar between the two)?

Discussion:

- I think the whole discussion could be condensed to some degree; for example I don’t think the discussion around the cost of Carotid route of exposure on lines 430-432 is really necessary.

- I am not sure how the exponential model provided the best-fit deterministic model outlined in lines 361-364: “The exponential model had the best performance during the evaluation of the deterministic models, whereas the two-parameter log-logistic and the multiple logistic regression models could not be fitted to more than one route of exposure”, when reviewing S3 and S4 the exponential regression model produced more non-significant parameters than the multiple regression model column for example

- I am not sure what the sentence “likely reflects an artefact of differential experimental data at lower doses” refers to (line 379), given that there were very limited number of studies using low infectious doses and I do not understand why these experiments would be different?

- References Jansen et al. 2021 (systematic review of taenia spp egg viability in different conditions & dispersal mechanisms) would be useful to support this point “Other sources of eggs such as the soil, water, or vectors with eggs would be necessary to study [45].” (line 403-404)

Jansen, F., Dorny, P., Gabriël, S. et al. The survival and dispersal of Taenia eggs in the environment: what are the implications for transmission? A systematic review. Parasites Vectors 14, 88 (2021). https://doi.org/10.1186/s13071-021-04589-6

- the point “could be expanded to explore the correlation between the inoculated dose and the number of cysts in pigs’ bodies and brains, with the purpose of further describing the disease” (lines 463 – 465) could be expanded to discuss linking infectious doses to understanding the population distribution of cysts (i.e. overdispersed distribution) in the pig host

Reviewer #2: The authors make a nice effort to test the best fitting models to dose response of Taenia solium so that they could be used in further transmission studies involving experimental infections and/or aiming at control measures with purpose of farm biosecurity or elimination of T.solium taeniasis/cysticercosis. In this effort, not only infection data from the literature was gathered, but also the authors added extra cysticercosis challenge data which has not yet been published. They used the data to find the best models and models' parameters to be used in future studies involving dose response for cysticercosis outcomes.

I recommend a mayor revision because there is a mayor issue that needs to be addressed, yet which i'll highlight in the comments.

My comments are as follow:

- The abstract needs, on top of anything written yet, a small line on the motivation to study T.solium (e.g., a shorter version of what the authors wrote in lines 75-78) .

- The parameter estimation of the exact Beta-Poisson (here on exact-BP) models and from the approximate-BP model is vaguely explained in lines 236-238. The authors should deepen in how this was done. It seems that they after finding parameter priors by max.lik. estimates they performed Markov chain Montecarlo simulations? was this really a Bayesian method? with which software was this doen? probabily R, with JAGS, or STAN? I can understand why was this not done also with the other models (2-par logistic, multi logistic and exponential) as just the simpler analysis used shows that they are not really fit to this dose response problem, but at least it should be mentioned why.

- Mayor issue: Related to my comment just above on whether there is an Bayesian MCMC approach included, and then further, this goes for all analyses accross the manuscript. The standard deviation of the data points is shown in the plots, but only of the dose. Given that for some readings there are 3 out of 4 pigs responding positive, or 5 out of 8, etc, this gives a large uncertainty to the values calculated for probability of infection. Therefore it would be appropriate to mention how this was tackled. Was it, e.g., with a reading 4/5, was P(dose) calculated as

P(dose) = 4/5 (which does not account for the large uncertainty)

or

4 ~ Binomial( p=P(dose), n=5 ) which properly accounts for the uncertainty (remember 4 out of 5 pigs positive is not statistically the same as 400 out of 500 pig positive).

So just as in all figures, the standard deviation of the data is mentioned only in dose, but the largest uncertainty still lies in the probability or response and this needs to be addressed. The only way to do this properly is with a Bayesian approach.

- For the SSE and R^2 analises (lines 251-253), what was used for the case of both BP models?

Given that the parameters had posterior distributions, and the SSE and R^2 analises tables show one hard coeficient (not ranges), was it the median of the posteriors?

- line 263: I see in table S3 and S4 that several of the approximate-BP are comparable to those of the exact-BP model, sometimes even smaller (but again no ranges). The choice of the exact-BP model is fine, but i don't see it strongly better than the approximate-BP model, except at very low doses... maybe this should be the reason to choose for this one? please add a comment on this.

- In the figures, indicate if the value of ID50 is the median ID50.

- In line 300 you refere to a quantity near to a single egg. Please provide interpretation of what a "fractional" egg would mean. As we know, eggs are counted, not continuous, so is either one or two or theree or none, etc. This would ligthen up the reading for many readers.

- All figures need to have larger font in the axes. They also need to have larger resolution (even better, use vector graphics). Actually in the figures included for the review, you just cannot read the numbers (you can deduce them though, so i still could review the manuscript).

- lines 342-343, maybe note that this also corresponds with the DR to viable cysts that las a lower dose for ID90-95.

-line 359 "deterministic models", exact- and approx-BP models are also deterministic, so just be clear by mentioning the other models (logistic, multilogistic and exponential)... Or am i missing something?

- lines 364-368: All these problems can be tackled within a Bayesian framework, but again, this would not make any fit for these models better, as you showed, and mentioned in line 369.

Reviewer #3: MAJOR COMMENTS

This is an interesting study that unfortunately suffers from major statistical deficiencies that preclude publication in its present state. I encourage revision to better articulate what can and cannot be done with the available data and what might be done to improve the design of future experiments. Studies failing to achieve their goals due to inadequately informative data are under-represented in the scientific literature and are an important part of the scientific record without which inadequately informative experiments may continue to be conducted and modelled.

In order to model P(cysts) as a function of dose, it is necessary for P(cysts) to change markedly between the tested doses (excluding negative controls). For most of the datasets considered, this is simply not the case as can be determined visually by plotting the empirical P(cysts) estimates (especially if Clopper-Pearson or Wilson score intervals are added). A likelihood ratio test can be performed to formally assess variation in response with respect to dose. If the increase in likelihood of the unpooled data with Pi = Xi/Ni is insignificant in comparison to the pooled data with P=SUM(Xi)/SUM(Ni), then the data may be pooled and there is no significant effect of the tested doses. [To fit a two-parameter model to such data is akin to fitting a two-parameter model to one datum, and will not provide meaningful extrapolation to a wider range of doses such as ID01 and ID50]. The only datasets for which it was possible to reject the null hypothesis (and thus conclude that dose has an effect) with a p-value <0.10 were 1) eggs with viable cysts, 2) eggs with brain cysts, and 3) beetles with viable cysts. [The latter of these is due to the low number of pigs with viable cysts at a dose of four beetles, which opposes an increase of dose with response and therefore negates all of the considered models]. Thus, it is my conclusion that dose-response modelling is only valid for the eggs data with viable cysts and brain cysts. [Based purely on subjective consideration of the data, the eggs data with any cysts might be worth an attempt at modelling despite the failed likelihood ratio test]. I suggest retaining the problematic data, discussing why modelling is not possible with these data, and describing how to improve the experimental design.

One of the issues with the proglottid and beetle data is that inherent clustering of eggs in the doses negates the Poisson assumption that is foundational to the exponential and exact beta-Poisson models (and the approximate beta-Poisson model when the approximation is valid for its mechanistic origins). This mechanistic flaw should be noted with discussion of what can be done to prevent it in experimental design and/or how to modify the models to account for such variation.

The multiple logistic regression model is potentially invalid because it assumes identical slope for each of the four dosing methods. This is an unnecessary restriction unless you add justification for it. Why not just carry out logistic regression on each dataset with unique slope and intercept, as is the case for log-logistic regression? That would be identical to the log-logistic regression but without the egg doses being log-transformed.

There is a general lack of clarity in the presented methodology that begs for provision of R scripts in the supplementary content to aid reproducibility. For example, a Bayesian method appears to have been used for fitting of the exact beta-Poisson model, but there is no discussion of the priors used, number of iterations, etc. to make the results reproducible.

MINOR COMMENTS

Lines 198-200 – The distinction between deterministic and stochastic models is unclear. All 5 are stochastic (they are all variants of binomial regression). The exponential and exact beta-Poisson models are mechanistic, the logistic and log-logistic models are not, and the approximate beta-Poisson model falls somewhere in between.

Lines 208-211 – Please specify the base of the logarithm. It is presumably 10, but “log” does not necessarily imply this (e.g., as the “log()” function returns a base 10 logarithm in Excel and a natural logarithm in R). It is not clear how the doses of zero are accommodated in the log-logistic model. Given that no cysts were detected, it would be reasonable to omit these data from the analysis as negative controls. [In all other models, inclusion or exclusion of the zero-dose group is irrelevant because the probability of detection is necessarily zero and the probability of no cysts is therefore necessarily 1].

Lines 228-232 – This sentence is incorrect because all three stated assumptions also apply to the exponential dose-response model. The presentation stops short of noting that the exact beta-Poisson model is a generalization of the exponential model.

Lines 242-247 – This approximation is only valid for beta>>alpha and beta>>1 (Teunis & Havelaar, 2000).

[I have only skimmed the remaining content with occasional notes]

Line 307 – It is not reasonable to estimate an ID50 that is nearly 40 orders of magnitude away from the nearest empirical data. This is a grievous error in extrapolation with a model that is poorly informed by the available data.

Lines 376-377 – As discussed in Schmidt (2015), a plateau in the probability of infection below 100% (possibly due to sterile immunity) causes the exact beta-Poisson parameters to approach zero and the probability of infection at low doses to be very high. The result is likely spurious and there are insufficient low-dose data to refute it.

Lines 450-454 – This is an interesting discussion. The dose-response experiments I have studied typically involve preparation of aliquots from a single well-mixed source so that the pathogens do not vary among doses. If the sources are inconsistent (e.g., such that one proglottid has viable eggs and the other does not), this could be a cause of flattening below 100% that is not due to sterile immunity of the pig.

REFERENCES

Schmidt (2015) - https://doi.org/10.1111/risa.12323

6. PLOS authors have the option to publish the peer review history of their article (what does this mean?). If published, this will include your full peer review and any attached files.

Reviewer #1: No

Reviewer #2: No

Reviewer #3: **Yes: **Philip J. Schmidt

---

## [Author Response · Author response to Decision Letter 0]

29 Oct 2021

REVIEWER #1

Intro

• Can the authors be more specific with the phrase “with heavy economic consequences” on line 99.

Thank you for letting us notice the lack of information. We have specified the idea in the following way:

Line 99: “…with heavy economic consequences such as an annual median loss of US $ 185.14 million in India, as estimated in 2011 [6].”

• The authors suggest the route of exposure for pigs is via ingestion of egg/proglottids in stools, however exposure may occur through consumption of eggs that have been dispersed in the environment (not in stools) via mechanical vectors. This is also mentioned as a “route of exposure” in the experimental studies in the methods (Table 1), so providing some context in the intro would be useful. See 

Jansen, F., Dorny, P., Gabriël, S. et al. The survival and dispersal of Taenia eggs in the environment: what are the implications for transmission? A systematic review. Parasites Vectors 14, 88 (2021). https://doi.org/10.1186/s13071-021-04589-6

We have included in the manuscript:

Line 106: “...pigs ingest T. solium eggs/proglottids in human feces or in the surrounded contaminated environment and develop ….”

Thank you very much for the paper of Jansen et al, 2021, an interesting piece of information, we included this as a citation.

Methods:

• Lines 128 onwards: It is not clear whether the search to identify cysticercosis challenge study data was conducted in a structured way (i.e., with a systematic search methodology) or non-structured? If structured, a flow diagram to identify the number of studies identified at each stage (i.e., title, abstract, full-text using the PRISMA method) should be included. If non-structured, I would suggest this literature search should be re-conducted using a systematic methodology, to highlight search terms/ literature databases so this search can be replicated (or updated in the future), and identify whether other selection criteria including language exclusion criteria etc to avoid reporting biases.

Based on the suggestion from Reviewer 1, we developed a systematic review; and included a detailed methodology to describe the method developed to include and exclude pigs in the model. This has enabled us to add one new paper to those we included in the study. We have also changed the wording and the structure of the paper to make this clear and follow the PRISMA method. A summary of the number of papers reviewed and methodology is now described in a new Figure 1.

• Can the authors explicitly say whether the criteria for pig selection (lines130 – 139) were used for pig selection in the CWG unpublished study (line 153) – I get this is true from the first paragraph in the methods, but think a few words on line 153 to this effect would improve clarity.

Thanks for letting us notice this point. We include the following sentence to explicitly say that the same criteria were used in the Methods section:

Line 162: “Additionally, thesis research and unpublished data from the Cysticercosis Working Group in Peru that were not found with this search strategy but matched the selection criteria after manual search were also included.”

• I do not understand why the number of exposed pigs varies so much between inoculum dose for each route of exposure in Table 2 CWG challenge study? Are there any sample size calculations behind the number of exposed pigs?

This is an interesting observation. The data comes from different experiments: Table 2 shows grouped data – this explains part of the variability. Further, the viability of eggs unfortunately varies considerably between tapeworms and within the tapeworm, as do the number and distribution of eggs in the inoculum. It is also known that the distribution of cysts load varies a lot in experimental infections with the same dose. All this may also have contributed to differences in the choice of sample size between the selected studies.

Besides, we added the following sentence regarding the limitations of the study:

Line 450: “The main limitations in this study were the limited number of data points and range of doses to develop a robust model. Further, the low sample sizes in some of the studies led to an increase of the uncertainty in the corresponding sections of the curve.”

• Table S1 refers to the route-dose-exposure-infection numbers from studies collated (and combined) from the literature (i.e. not including CWG unpublished study)? It is not clear in the methods text (line 160).

Actually, the unpublished studies from the CWGP are included in Table S1 as UD (Unpublished Data). Because of the current structure of the manuscript, the idea of the method text is located at the Results section, and we specify that the unpublished studies are included in the following way:

Line 283: “A table listing the dose groups with their respective reference (both published and unpublished studies) can be found in S1 Table.”

• I do not think line 178-183 are necessary if this is a secondary analysis of published studies, but in the case of the CWG unpublished study there should be reference to ethical approval only.

We also consider that, given that this is a secondary analysis, ethical approval is not necessary. If the journal agrees we can eliminate that paragraph and include the ethical approvals for unpublished CWG data in the Results section as:

Line 302: “The unpublished study conducted by the Cysticercosis Working Group in Peru included in this paper was reviewed and approved by the Institutional Ethics Committee for the Use of Animals (Comite de Ética y Bienestar Animal (CEBA)) at the School of Veterinary Medicine, Universidad Nacional Mayor de San Marcos (protocol no. 2009-003, 2010-012, and 2012-004), and by the Institutional Ethics Committee for the Use of Animals (Comité Institucional de Ética para el Uso de Animales (CIEA)) at the Universidad Peruana Cayetano Heredia (protocol no. A5146-01), both located in Lima, Peru.”

• Was the distribution for eggs from beetles a log-normal distribution to reflect the skewedness (it would be useful for the authors to be more explicit here on lines 191-192)

The distribution is more like a negative binomial, but due to the lack of enough data we preferred not to use parameters for a negative binomial, and we used a random sampling. To be more explicit, we modify the text as:

Line 188: “For the “Beetles” pathway, we used the database from Gomez-Puerta et al. [37] to estimate the dose for each beetle by bootstrapping (resampling). The database comprises 35 beetles, with a median of 21 eggs, a range of 3 to 235 eggs per beetle, and a right-skewed distribution (negative binomial distribution).”

• What was the criteria for determining that the five dose-response models could produce a dose-response curve fitting (was this just a visual check, or any other diagnostics used?) on lines 249-251?

Thanks for letting us notice the lack of information. The criteria used to determine that the models could produce a dose-response curve were visual checking for all the models and the estimation of significant parameters only for the two-parameter log-logistic, logistic regression, and exponential regression models. Besides, we added the following sentence:

Line 251: “Whether the different types of dose-response models could be used to produce a dose-response curve fitting relevant exposure pathways was first assessed by visual checking (for all models) then by estimating the value of model parameters. While this was done for all models, more attention was paid to estimating the parameters for models that, visually, seemed to have a chance of providing a good fit for the data.”

• I do not see the probability of surviving the chain of barriers to infection (r) on lines 233-234 in equation 5, or perhaps I am misunderstanding this function?

The idea in this paragraph is that the probability of surviving the chain of barriers to infection follows a Beta distribution through the inclusion of the Kummer confluent hypergeometric function (1F1). This function replaces the (r) in equation 5. In order to avoid any type of ambiguity, we decided to remove “(r)” from this paragraph (line 234).

• Are there any reasons for selecting minimum doses required to produce a probability of infection of 1% and 50% (and not including higher probabilities i.e. probability of infection of 95%?) on lines 253-255.

First, thank you for letting us notice the mistake, we removed the word “minimum”, as the estimation was for the median as is detailed in Table 4 where it is also mentioned. Dose-response studies usually present mainly the 50% probability of infection but in some there is also information about 95% probability. However, in our case, the 95% probability is out of the range of observations for most of the curves, leading to a higher uncertainty and possibly misleading interpretations. We therefore considered only the 1% and 50% probabilities because these allow us to know the performance of the dose-response curve with more accuracy rather than evaluating higher doses.

• Can the authors provide further details on how the three oral routes of exposure were merged into the single “oral” pathway in lines 256-257?

In this case, the data was analyzed together, using all these routes and our knowledge of the approximate number of eggs per beetle or proglottid to build the model. We have modified the paragraph in order to clarify this topic:

Line 262: “Finally, the data from all three oral exposure pathways (i.e., “Proglottids”, “Eggs” and “Beetles”) were analyzed together to build a single “Oral” pathway with the purpose of obtaining a more robust model with a wider range of observations reflecting the plausible natural infection pathways of the disease.”

Results:

• So I am clear, the beta-Poisson model was found to best fit the data for all exposure routes in lines 262-263

Yes, this is correct.

• Can the authors combine supplementary tables S3 and S4 comparing model fits with either SEE or R2 to facilitate an easier comparison?

The supplementary tables S3 and S4 have been combined in “Appendix S1. Assessment of the fit of the dose-response models”.

• The axis numbers (especially on the x-axis) of figures 1 – 4 are very hard to see without zooming in substantially, please could these be increased in size.

The size of the X axes was increased.

• I am unclear how the statement “For the development of any type of cyst, the four routes of infection presented a high probability of infection at low doses (Table 4)” (lines 297 – 300) can be obtained from table 4, as this only looks 1 and 50% probabilities of infection (without referring to the figures?), and equally the next sentence seems to suggest this can be read from Table 4, but can only really be read from Figures 1-3 if I am understanding correctly, so I think to improve clarity it would be useful to show these sentences indicate when referring to the figures (throughout lines 297 – 309)?

Thanks for the observation. We reviewed the wording to explain that the new Table 4 shows that low doses are sufficient to obtain a 1% or 50% probability of infection, while new Figures 2 and 3 show that even a probability of infection of 80% can be obtained through relatively low doses. 

• I do not understand where the “half the inoculated pigs” comes from in the “to obtain viable cysts in at least half of the inoculated pigs” (lines 304-303); is this what ID50 is measuring in terms of the probability of infection in at least 50% of the exposed pigs? (Although I thought this was the minimum dose to obtain a 50% probability of infection in any pig)?

Thanks for letting us notice the mistake. The ID50 is measuring the dose required to obtain cysts in one pig with a 50% probability. However, this sentence was removed from the manuscript due to the new structure of the Results section as a recommendation of Reviewer 3.

• I am not sure what the phrase “notorious reduction” means on line 306

The phrase was removed from the manuscript due to the new wording of the Results section.

• I do not understand what the sentence “even though one of these routes of exposure bypasses the mouth (“Eggs”)” (line 322)” – I would have thought the eggs are only consumed by the pigs via the mouth (unless the authors are talking about the Carotid inoculation pathway?)

Eggs were inoculated through an esophageal tube, that’s why we said that this route bypasses the oral cavity. The procedure was also described in Table 1. Besides, the paragraph which contained this sentence was removed due to the new structure of the Results section.

• However the ID50 infectious dose is >100 for any cyst in the oral pathway (4A) but <100 for any cyst in the carotid pathway, which is an important distinction re lines 337-339, so I am not sure the median probabilities are so comparable (although the overall shape is similar between the two)?

Due to Reviewer 3’s recommendations, we are now only comparing the “Oral” and “Eggs” pathways. The “Carotid” pathway was moved to the S2 Appendix and is no longer considered in the comparison within the manuscript.

Discussion:

• I think the whole discussion could be condensed to some degree; for example I don’t think the discussion around the cost of Carotid route of exposure on lines 430-432 is really necessary.

The discussion was condensed in function of the new structure of the manuscript.

• I am not sure how the exponential model provided the best-fit deterministic model outlined in lines 361-364: “The exponential model had the best performance during the evaluation of the deterministic models, whereas the two-parameter log-logistic and the multiple logistic regression models could not be fitted to more than one route of exposure”, when reviewing S3 and S4 the exponential regression model produced more non-significant parameters than the multiple regression model column for example

Although the multiple logistic regression model presented a greater number of curves with significant parameters than the exponential model, the function of this model is not adjusted to present a probability of 0% when the dose is 0 as in the exponential model. Besides, the multiple logistic regression was replaced by the simple logistic regression due to the recommendation from Reviewer 3; however, a similar performance than the multiple regression model was observed.

• I am not sure what the sentence “likely reflects an artefact of differential experimental data at lower doses” refers to (line 379), given that there were very limited number of studies using low infectious doses and I do not understand why these experiments would be different?

What we wanted to emphasize here is that given the lack of information available regarding infection with lower doses, the reader may falsely assume that the infection at lower doses is different between the egg model and the others. This is clarified in Appendix S2 now. We did not put more comment in the main text as, following comments from another reviewer, we are now focusing on the “Eggs” and “Oral” pathways only.

• References Jansen et al. 2021 (systematic review of taenia spp egg viability in different conditions & dispersal mechanisms) would be useful to support this point “Other sources of eggs such as the soil, water, or vectors with eggs would be necessary to study [45].” (line 403-404)

Jansen, F., Dorny, P., Gabriël, S. et al. The survival and dispersal of Taenia eggs in the environment: what are the implications for transmission? A systematic review. Parasites Vectors 14, 88 (2021). https://doi.org/10.1186/s13071-021-04589-6

Thank you very much for the suggestion, the citation was included (line 430).

• the point “could be expanded to explore the correlation between the inoculated dose and the number of cysts in pigs’ bodies and brains, with the purpose of further describing the disease” (lines 463 – 465) could be expanded to discuss linking infectious doses to understanding the population distribution of cysts (i.e. overdispersed distribution) in the pig host

We added the following phrase in the manuscript:

Line 466: “Further, in future work, the current framework could be expanded to explore the correlation between the inoculated dose and the number of cysts in pigs’ bodies and brains and even link the infectious dose to the cyst distribution in the pig, with the purpose of further describing the disease.”

REVIEWER #2

• The abstract needs, on top of anything written yet, a small line on the motivation to study T.solium (e.g., a shorter version of what the authors wrote in lines 75-78).

Thanks for the advice. We added the following phrase at the beginning of the Abstract:

Line 47: “Taenia solium is an important cause of acquired epilepsy worldwide and remains endemic in Asia, Africa, and Latin America.”

• The parameter estimation of the exact Beta-Poisson (here on exact-BP) models and from the approximate-BP model is vaguely explained in lines 236-238. The authors should deepen in how this was done. It seems that they after finding parameter priors by max.lik. estimates they performed Markov chain Montecarlo simulations? was this really a Bayesian method? with which software was this doen? probabily R, with JAGS, or STAN? I can understand why was this not done also with the other models (2-par logistic, multi logistic and exponential) as just the simpler analysis used shows that they are not really fit to this dose response problem, but at least it should be mentioned why.

Thanks for letting us notice the lack of information. The parameter estimation of both beta-Poisson models was carried out in R v. 4.1.1 and is specified at the beginning of the “Dose-response analysis” section of Methods. Furthermore, we specified the followed framework for the BP models in the following way:

Line 236: “The parameters were estimated following Xie et al.’s approach [45], which consisted in estimating the parameters by maximum likelihood and using them as prior information to generate, by a bootstrap algorithm of 5000 iterations, a sample-size-dependent confidence interval for each parameter in the model.

• Mayor issue: Related to my comment just above on whether there is an Bayesian MCMC approach included, and then further, this goes for all analyses accross the manuscript. The standard deviation of the data points is shown in the plots, but only of the dose. Given that for some readings there are 3 out of 4 pigs responding positive, or 5 out of 8, etc, this gives a large uncertainty to the values calculated for probability of infection. Therefore it would be appropriate to mention how this was tackled. Was it, e.g., with a reading 4/5, was P(dose) calculated as

P(dose) = 4/5 (which does not account for the large uncertainty)

or

4 ~ Binomial( p=P(dose), n=5 ) which properly accounts for the uncertainty (remember 4 out of 5 pigs positive is not statistically the same as 400 out of 500 pig positive).

So just as in all figures, the standard deviation of the data is mentioned only in dose, but the largest uncertainty still lies in the probability or response and this needs to be addressed. The only way to do this properly is with a Bayesian approach.

The approach employed in both beta-Poisson models (approximate and exact) is based in a bootstrap algorithm which gives a sample-size dependent confidence interval. A reduced sample size in the data is reflected in the high uncertainty of the resulting model in the sections of the curve where the observed data is composed by few infected pigs.

• For the SSE and R^2 analises (lines 251-253), what was used for the case of both BP models?

Given that the parameters had posterior distributions, and the SSE and R^2 analises tables show one hard coeficient (not ranges), was it the median of the posteriors?

To determine the sum of squared errors of prediction (SSE) and the coefficient of determination (R2) of the beta-Poisson models (both approximate and exact), only the median probability of infection was used in the calculation. In order to avoid any type of misunderstanding, we added the following sentence:

Line 255: “Subsequently, each dose-response curve was analyzed through the calculation of both the residual sum of squared errors of prediction (SSE) and the coefficient of determination (R2) to determine the type of model that best fits the data. For both beta-Poisson models, only the median probability of infection was considered to develop these calculations.”

• line 263: I see in table S3 and S4 that several of the approximate-BP are comparable to those of the exact-BP model, sometimes even smaller (but again no ranges). The choice of the exact-BP model is fine, but i don't see it strongly better than the approximate-BP model, except at very low doses... maybe this should be the reason to choose for this one? please add a comment on this.

Thanks for letting us notice this lack of information. The choice of the exact-BP rather than the approximate-BP was because the approximate-BP did not fulfill in most of the cases the two premises to be a valid model: (1) the beta parameter (β) must be greater than the alpha parameter (α), and (2) the beta parameter must be greater than 1. To clarify this, we added the following phrase in the Methods section when explaining about the approximate-BP model:

Line 246: “However, this approximation is only valid when two premises are fulfilled: β » 1, and α « β [42].”

And added the following phrase in the Results section:

Line 328: “Moreover, with the approximate beta-Poisson model, we found parameters similar to those of the exact beta-Poisson model, but they did not fulfill the premises previously mentioned (β » 1 and α « β) in most of the dose-response curves and presented misleading results at the beginning of the curve as shown in Appendix S6.”

• In the figures, indicate if the value of ID50 is the median ID50.

We have added “median ID50” in all the figures.

• In line 300 you refere to a quantity near to a single egg. Please provide interpretation of what a "fractional" egg would mean. As we know, eggs are counted, not continuous, so is either one or two or theree or none, etc. This would ligthen up the reading for many readers.

To clarify this, we consulted to Dr. Gertjam Medema, who is a renowned expert in this area, below his response:

“When it comes to infection probabilities, you should look from the population perspective, not from the individual’s perspective. Your ID5 will be around 0.1 eggs and your ID1 around 0.03 eggs. As the concentration of eggs goes down, you will not infect everybody with an egg, but only a small fraction of the population with 1 (or maybe 2) and the rest with 0 eggs. You will see that if you plot the Pinf on a log scale you will get a straight line to the lower concentrations. Your Pinf at a dose of 1 egg is around 30%, your Pinf at 0.1 eggs (or 1 egg in 10 persons) around 3%, 0.01 eggs (1 egg in 100 persons) around 0.3%, 0.001 eggs around 0.03% and so on.”

Therefore, we added the following sentence in the manuscript:

Line 343: “Finally, please note that some of the computations give infective doses below one egg. What a fractional dose of 0.056 eggs to reach ID01 means is that giving one egg to each pig in a sample would lead to an expected share of infected pigs higher than one percent. To infect 10 pigs out of a sample of 1000 (share of 1%), for example, one may need to give one egg to 56 pigs and none to the other 944 pigs.”

• All figures need to have larger font in the axes. They also need to have larger resolution (even better, use vector graphics). Actually in the figures included for the review, you just cannot read the numbers (you can deduce them though, so i still could review the manuscript).

We improved the size of the axes and the resolution of the figures.

• lines 342-343, maybe note that this also corresponds with the DR to viable cysts that las a lower dose for ID90-95.

It is an interesting interpretation. Although the viable brain cysts are considered in the probability for the development of viable cyst, the pigs that develop brain cysts are already infected with a high concentration of cysts in the carcass (including viable cysts); therefore, we do not believe that the brain viable cysts could have that kind of impact over the dose-response relationship for the development of viable cysts.

• line 359 "deterministic models", exact- and approx-BP models are also deterministic, so just be clear by mentioning the other models (logistic, multilogistic and exponential)... Or am i missing something?

We understand that a deterministic model is a model in which no randomness is involved in the development of future states of the system. A stochastic model has a random probability distribution or pattern that may be analyzed statistically but may not be predicted precisely.

• lines 364-368: All these problems can be tackled within a Bayesian framework, but again, this would not make any fit for these models better, as you showed, and mentioned in line 369.

The Bayesian framework were applied during the dose-response analysis using the approximate and exact beta-Poisson model. Besides, a Bayesian framework using a model with a function that is adjusted to initiate with a probability of 0% can also help to address these problems as seen with the exact beta-Poisson model.

REVIEWER #3

MAJOR COMMENTS

• This is an interesting study that unfortunately suffers from major statistical deficiencies that preclude publication in its present state. I encourage revision to better articulate what can and cannot be done with the available data and what might be done to improve the design of future experiments. Studies failing to achieve their goals due to inadequately informative data are under-represented in the scientific literature and are an important part of the scientific record without which inadequately informative experiments may continue to be conducted and modelled.

In order to model P(cysts) as a function of dose, it is necessary for P(cysts) to change markedly between the tested doses (excluding negative controls). For most of the datasets considered, this is simply not the case as can be determined visually by plotting the empirical P(cysts) estimates (especially if Clopper-Pearson or Wilson score intervals are added). A likelihood ratio test can be performed to formally assess variation in response with respect to dose. If the increase in likelihood of the unpooled data with Pi = Xi/Ni is insignificant in comparison to the pooled data with P=SUM(Xi)/SUM(Ni), then the data may be pooled and there is no significant effect of the tested doses. [To fit a two-parameter model to such data is akin to fitting a two-parameter model to one datum, and will not provide meaningful extrapolation to a wider range of doses such as ID01 and ID50]. The only datasets for which it was possible to reject the null hypothesis (and thus conclude that dose has an effect) with a p-value <0.10 were 1) eggs with viable cysts, 2) eggs with brain cysts, and 3) beetles with viable cysts. [The latter of these is due to the low number of pigs with viable cysts at a dose of four beetles, which opposes an increase of dose with response and therefore negates all of the considered models]. Thus, it is my conclusion that dose-response modelling is only valid for the eggs data with viable cysts and brain cysts. [Based purely on subjective consideration of the data, the eggs data with any cysts might be worth an attempt at modelling despite the failed likelihood ratio test]. I suggest retaining the problematic data, discussing why modelling is not possible with these data, and describing how to improve the experimental design.

Dear Dr Schmidt, 

We are very grateful for your comments, and your observations have been very useful for the development of the manuscript. We consider that it is necessary to give instructions in this paper to collect better information that allows the development of better dose-response models. As you mention, this does not mean that the models that can be developed with the information available should not be published; this has been done with models for other microorganisms. We have made the suggested changes based on the data that can best inform a preliminary model and we have clarified in the manuscript the limitations that these models have. We have included new guidelines for future experimental infections that can feed models with better data.

• One of the issues with the proglottid and beetle data is that inherent clustering of eggs in the doses negates the Poisson assumption that is foundational to the exponential and exact beta-Poisson models (and the approximate beta-Poisson model when the approximation is valid for its mechanistic origins). This mechanistic flaw should be noted with discussion of what can be done to prevent it in experimental design and/or how to modify the models to account for such variation.

We don’t believe this has much of an impact on the results. Although proglottids appear to be a package of eggs, they will easily be disaggregated in the stomach of the pig, allowing the contact with mucosal surface and entry into the blood stream. In support to that interpretation, eggs are always found disaggregated in coprological analysis, contrary to what can be seen with viruses where there is microscopical evidence of aggregation. For that reason, we consider the Poisson model appropriate.

• The multiple logistic regression model is potentially invalid because it assumes identical slope for each of the four dosing methods. This is an unnecessary restriction unless you add justification for it. Why not just carry out logistic regression on each dataset with unique slope and intercept, as is the case for log-logistic regression? That would be identical to the log-logistic regression but without the egg doses being log-transformed.

The results of the multiple logistic regression were replaced for the parameters of the logistic regression on each dataset with unique slope and intercept as suggested.

• There is a general lack of clarity in the presented methodology that begs for provision of R scripts in the supplementary content to aid reproducibility. For example, a Bayesian method appears to have been used for fitting of the exact beta-Poisson model, but there is no discussion of the priors used, number of iterations, etc. to make the results reproducible.

Thanks for letting us notice the lack of information. We have uploaded the R scripts to a repository in GitHub and specified it at the beginning of the “Dose-response analysis” section in Results:

Line 319: “The R scripts to reproduce the results of the dose-response analysis can be found in the following repository in GitHub: https://github.com/dandradem/cysticercosis--dose-response.”

Besides, we specified the followed framework for the beta-Poisson models in the following way:

Line 236: “The parameters were estimated following Xie et al.’s approach [45], which consisted in estimating the parameters by maximum likelihood and using them as prior information to generate, by a bootstrap algorithm of 5000 iterations, a sample-size-dependent confidence interval for each parameter in the model.”

MINOR COMMENTS

• Lines 198-200 – The distinction between deterministic and stochastic models is unclear. All 5 are stochastic (they are all variants of binomial regression). The exponential and exact beta-Poisson models are mechanistic, the logistic and log-logistic models are not, and the approximate beta-Poisson model falls somewhere in between.

We understand that a deterministic system is a system in which no randomness is involved in the development of future states of the system. A stochastic system has a random probability distribution or pattern that may be analyzed statistically but may not be predicted precisely. In order to avoid misunderstanding with technical terms that do not affect the interpretation of the paper, we decide not to use the terms.

• Lines 208-211 – Please specify the base of the logarithm. It is presumably 10, but “log” does not necessarily imply this (e.g., as the “log()” function returns a base 10 logarithm in Excel and a natural logarithm in R). It is not clear how the doses of zero are accommodated in the log-logistic model. Given that no cysts were detected, it would be reasonable to omit these data from the analysis as negative controls. [In all other models, inclusion or exclusion of the zero-dose group is irrelevant because the probability of detection is necessarily zero and the probability of no cysts is therefore necessarily 1].

The base of the natural logarithm is specified in the Equation 1 (line 211).

Regarding the log-logistic model, the inclusion of the negative controls in the analysis did not represent an obstacle to obtain parameters despite these being non-significant. The cases where it was not possible to obtain parameters were when the few data collected with high-dose levels provided null information about the rising of the curve.

• Lines 228-232 – This sentence is incorrect because all three stated assumptions also apply to the exponential dose-response model. The presentation stops short of noting that the exact beta-Poisson model is a generalization of the exponential model.

We have modified the manuscript to clarify this misunderstanding, as follows:

Line 225: “This model, as well as the beta-Poisson models described below, makes the following three assumptions [44]: only one viable organism is required to produce the infection process, the exact number of organisms inoculated in each dose follows a Poisson distribution, and the survival of any organism in a single host is independent from the survival of other organisms in that host.”

• Lines 242-247 – This approximation is only valid for beta>>alpha and beta>>1 (Teunis & Havelaar, 2000).

Thanks for letting us notice this lack of information, we added the following phrase in the Methods section when explaining about the approximate-BP model:

Line 246: “However, this approximation is only valid when two premises are fulfilled: β » 1, and α « β [42].”

And added the following phrase in the Results section:

Line 328: “Moreover, with the approximate beta-Poisson model, we found parameters similar to those of the exact beta-Poisson model, but they did not fulfill the premises previously mentioned (β » 1 and α « β) in most of the dose-response curves and presented misleading results at the beginning of the curve as shown in Appendix S6.”

[I have only skimmed the remaining content with occasional notes]

• Line 307 – It is not reasonable to estimate an ID50 that is nearly 40 orders of magnitude away from the nearest empirical data. This is a grievous error in extrapolation with a model that is poorly informed by the available data.

Due to the recommendation of focusing the paper on the presentation of the “Eggs” pathway, the results of the other pathways (where the ID50 that was nearly 40 orders of magnitude away) were moved to Supplementary Information (S2 Appendix). The ID50 is still included with the purpose of describing the performance of the dose-response curve.

• Lines 376-377 – As discussed in Schmidt (2015), a plateau in the probability of infection below 100% (possibly due to sterile immunity) causes the exact beta-Poisson parameters to approach zero and the probability of infection at low doses to be very high. The result is likely spurious and there are insufficient low-dose data to refute it.

We have included in the manuscript: 

Line 394: “As compared to the other pathways, the “Eggs” pathway was informed by a broader range of experimental data, including a number of experiments with small doses of eggs, with 10 eggs leading to successful infection in some but not all of the pigs. This resulted in a flattening of the slope of the curve at the lowest estimated doses as compared to what could have been obtained with other pathways.”

Besides, we have added the following idea when talking about the “Proglottids”, “Beetles”, and “Carotid” pathways:

Line 390: “Another issue regarding the development of dose response curves is when the probability of infection is below 100% at high doses, corresponding to an immunity plateau and an overestimation of the probability of infection at low doses (Appendix S2) [48]. Consequently, the inclusion of an immune parameter in the exact beta-Poisson function is highly recommended for future dose-response analysis to obtain best-fitted models.”

And the following when talking about the “Eggs” pathway:

Line 398: “Besides, this pathway presented observed data with probabilities of infection for the development of both any and viable cysts equal to 100% at high doses. Therefore, we believe that the dose-response curve developed for the “Eggs” pathway can estimate the probability of infection at lower doses in a reasonably realistic way as compared to other pathways, even without the inclusion of additional parameters.” 

Thank you very much for the paper of Schmidt, 2015, an interesting piece of information. We included this as a citation.

• Lines 450-454 – This is an interesting discussion. The dose-response experiments I have studied typically involve preparation of aliquots from a single well-mixed source so that the pathogens do not vary among doses. If the sources are inconsistent (e.g., such that one proglottid has viable eggs and the other does not), this could be a cause of flattening below 100% that is not due to sterile immunity of the pig.

REFERENCES

Schmidt (2015) - https://doi.org/10.1111/risa.12323

We would like to thank Dr. Schmidt a lot for so interesting comments and suggestions that have help a lot and allows us to have a better understanding of dose-response development area. We considered that more information should be available in the literature to use and improve experimental infection data that would be used to enlarge sample size of this kind of analysis.

---

## [Decision Letter · Decision Letter 1]

20 Dec 2021

PONE-D-21-23111R1Development of a dose-response model for porcine cysticercosisPLOS ONE

Dear Dr. Gonzales-Gustavson,

Thank you for submitting your manuscript to PLOS ONE. After careful consideration, we feel that it has merit but does not fully meet PLOS ONE’s publication criteria as it currently stands. Therefore, we invite you to submit a revised version of the manuscript that addresses the points raised during the review process.

We look forward to receiving your revised manuscript.

Kind regards,

Brecht Devleesschauwer

Academic Editor

PLOS ONE

Journal Requirements:

Additional Editor Comments:

**All reviewers appreciated the revisions made to the manuscript. Reviewer #3 raised some further issues which could be addressed in a final, minor revision round.**

Reviewers' comments:

Reviewer's Responses to Questions

**Comments to the Author**

1. If the authors have adequately addressed your comments raised in a previous round of review and you feel that this manuscript is now acceptable for publication, you may indicate that here to bypass the “Comments to the Author” section, enter your conflict of interest statement in the “Confidential to Editor” section, and submit your "Accept" recommendation.

Reviewer #2: All comments have been addressed

Reviewer #3: All comments have been addressed

2. Is the manuscript technically sound, and do the data support the conclusions?

Reviewer #2: Yes

Reviewer #3: Yes

3. Has the statistical analysis been performed appropriately and rigorously? 

Reviewer #2: Yes

Reviewer #3: (No Response)

4. Have the authors made all data underlying the findings in their manuscript fully available?

Reviewer #2: Yes

Reviewer #3: Yes

5. Is the manuscript presented in an intelligible fashion and written in standard English?

Reviewer #2: Yes

Reviewer #3: Yes

6. Review Comments to the Author

Reviewer #2: The authors have addressed satisfactorily all points I raised in my previous review, with good explanations to me and by improving the manuscript accordingly. I'm pleased to say Good job :) !

Reviewer #3: MINOR COMMENTS

It isn’t clear how relevant non-viable cysts and brain cysts are. Are pig brains ingested by humans? Are non-viable cysts of concern to the health of the human or pig in some way? It seems like more context on the transmission process would be helpful, though perhaps not strictly necessary.

Lines 53-54 – I suggest “…with ingestion of proglottids, eggs, and beetles that ingested eggs, and direct injection of activated oncospheres into the carotid artery” or similar. As written, “injected directly into the carotid artery” seems to apply to all four types of dosed material.

Lines 139-140 – I suggest “…a systemic infection results from a single infective dose…”

Lines 186-191 – It is not very clear how these conversions from proglottid/beetle doses to egg doses were implemented.

Line 195 – Table S2 is cross-referenced in the text before Table S1. Please check cross-referencing of tables and appendices throughout to make sure it is correct, complete, and numbered in order of appearance in the manuscript.

Line 201 – “…log-logistic regression, logistic regression, the exponential model, and approximate and exact beta-Poisson models”. “Exponential regression” is not quite right (here and throughout).

Line 207 – Add “regression”? Although it is reparameterized in terms of ID50, it is still a generalized linear model.

Line 211 – I believe that there is a missing negative sign in front of beta_slope

Line 220 – It isn’t meaningful to describe both Pinf and r as “probability of infection”. Perhaps r is the “probability of infection from exactly one infectious agent”.

Line 223 – Why not use maximum likelihood consistently for all models? Is the result the same?

Figure 1 – The resolution of this figure is very poor

Lines 255-259 – It isn’t clear how R^2 is being calculated for the non-regression models. I didn’t look at the R scripts closely enough to figure it out. If all models are fit by maximum likelihood, the Akaike information criterion is a simple model comparison tool for non-nested models such as these.

Line 316 – Pooling data in this way presumes that all three scenarios share the same dose-response model. It is possible to use a likelihood ratio test to determine whether unpooled analysis (e.g., a unique set of parameter values for each scenario) provides a significantly better fit than pooled analysis (e.g., having a shared set of parameter values common to all three scenarios). Alternatively, you could compare the AIC of the pooled data model with the AIC for the three models with unpooled data.

Table 3 – It is typical to report maximum likelihood estimates of parameters rather than medians.

Line 387 – Appendix S2

There is a very small number of grammatical issues throughout:

- Line 60: Revise to “…each of the four…”

- Line 77: Revise to “…models have been developed.” (to clarify that such models were not part of this work)

- Line 107: Revise to “surrounding”

- Lines 158-162: It is problematic to have two colons in this long sentence, and use “operators”.

- Lines 281, 283: Revise to “cyst”

7. PLOS authors have the option to publish the peer review history of their article (what does this mean?). If published, this will include your full peer review and any attached files.

Reviewer #2: No

Reviewer #3: **Yes: **Philip J. Schmidt

---

## [Author Response · Author response to Decision Letter 1]

1 Feb 2022

REVIEWER #2

• The authors have addressed satisfactorily all points I raised in my previous review, with good explanations to me and by improving the manuscript accordingly. I'm pleased to say Good job :) !

Thank you for your valuable recommendations. These have improved the manuscript in a significantly way.

REVIEWER #3

MINOR COMMENTS

• It isn’t clear how relevant non-viable cysts and brain cysts are. Are pig brains ingested by humans? Are non-viable cysts of concern to the health of the human or pig in some way? It seems like more context on the transmission process would be helpful, though perhaps not strictly necessary.

Regarding the degenerated (non-viable) cysts, their relevance depends on the term of the infection in pigs. In short-term, their importance lies in the purpose of the research that requires the development of experimental infection in pigs. In contrast, the degenerated cysts mostly disappears while a reduced number calcifies in the pigs’ carcass remaining uncapable of continuing the life cycle of the parasite.

To highlight the importance of the models for the development of brain cysts, we have added the following paragraph:

Line 453: “Despite the presence of cysts in the pigs’ brain represents a null impact on public health, the models for the development of brain cysts can also serve as reference to estimate the required doses to obtain cysts in pigs’ brain experimentally for future studies with the aim of evaluating treatment schemes for NCC using pigs as animal models. On the other hand, the model for development of brain cysts through the “Oral” pathway may be extrapolated to human NCC based on recent evidence which suggests that oncospheres are distributed to all tissues in humans, instead of being established preferentially in the brain, as was believed before [18]. However, it is necessary to demonstrate that there is no difference in the distribution of cysts between humans and pigs in order to carry out such extrapolation.”

• Lines 53-54 – I suggest “…with ingestion of proglottids, eggs, and beetles that ingested eggs, and direct injection of activated oncospheres into the carotid artery” or similar. As written, “injected directly into the carotid artery” seems to apply to all four types of dosed material.

As suggested, we have modified the text as:

Line 53: “…with ingestion of proglottids, eggs, and beetles that ingested eggs, and direct injection of activated oncospheres into the carotid artery.”

• Lines 139-140 – I suggest “…a systemic infection results from a single infective dose…”

We have modified the text as suggested in the following way:

Line 139: “…because a systemic infection results from a single infective dose…”

• Lines 186-191 – It is not very clear how these conversions from proglottid/beetle doses to egg doses were implemented.

Thanks for letting us notice the lack of information. We have modified the idea in the following way to avoid any type of misunderstanding:

Line 186: “In the studies that used proglottids, the infective dose for each proglottid was estimated by sampling with replacement a randomly-generated database composed of 10,000 observations following a discrete uniform distribution between 30,000-50,000 eggs for a full proglottid [13], and then linearly scaled for ½ and ¼ of a proglottid. For the “Beetles” pathway, we used the database composed by experimentally infected beetles with T. solium eggs from Gomez-Puerta et al. [37] to estimate the dose for each beetle by sampling with replacement.”

• Line 195 – Table S2 is cross-referenced in the text before Table S1. Please check cross-referencing of tables and appendices throughout to make sure it is correct, complete, and numbered in order of appearance in the manuscript.

We have reordered the supplementary information according to the order of appearance in the manuscript.

• Line 201 – “…log-logistic regression, logistic regression, the exponential model, and approximate and exact beta-Poisson models”. “Exponential regression” is not quite right (here and throughout).

As suggested, we have replaced the term “exponential regression” by “exponential model” both in the manuscript and in S5 Appendix.

• Line 207 – Add “regression”? Although it is reparameterized in terms of ID50, it is still a generalized linear model.

We have added the term “regression” both in the manuscript and in S3 Appendix as suggested.

• Line 211 – I believe that there is a missing negative sign in front of beta_slope

As seen in “Ritz C, Baty F, Streibig JC, Gerhard D. Dose-response analysis using R. PLoS One. 2015;10: 1–13. doi:10.1371/journal.pone.0146021” (reference of the R package used to perform the dose-response analysis with this model) referenced in this section, beta_slope does not have a negative sign in the log-logistic regression function.

• Line 220 – It isn’t meaningful to describe both Pinf and r as “probability of infection”. Perhaps r is the “probability of infection from exactly one infectious agent”.

Thanks for letting us notice this point. We have modified the idea in the following way to avoid misunderstandings:

Line 222: “This model has one parameter: the probability of at least one pathogen surviving the chain of barriers and causing an infection from any quantity of ingested pathogens (r) [42,43] (see Equation 3).”

• Line 223 – Why not use maximum likelihood consistently for all models? Is the result the same?

Granted that the exponential model presented in this study is a non-linear model, the relationship cannot be linearized by transformation of the response variable or the explanatory variable (or both). Therefore, the parameters of this model only can be estimated by non-linear least squares rather than by maximum likelihood estimation.

• Figure 1 – The resolution of this figure is very poor

The resolution of Figure 1 has been improved as requested according to the submission guidelines for figures.

• Lines 255-259 – It isn’t clear how R^2 is being calculated for the non-regression models. I didn’t look at the R scripts closely enough to figure it out. If all models are fit by maximum likelihood, the Akaike information criterion is a simple model comparison tool for non-nested models such as these.

We have modified the following sentence in order to avoid any type of misunderstanding:

Line 259: “For both beta-Poisson models, only the median probability of infection resulting in the posterior distribution was considered and compared to observed data to develop these calculations.”

Regarding the use of AIC, not all the models were fitted by MLE as can be seen in the Exponential model which was fitted by non-linear least squares due to not being a linear model. Besides, the coefficient of determination (R2) and the Sum of Squared Errors of Prediction (SSE) are valid calculations to estimate the variation between the predicted and the observed outcomes in these models.

• Line 316 – Pooling data in this way presumes that all three scenarios share the same dose-response model. It is possible to use a likelihood ratio test to determine whether unpooled analysis (e.g., a unique set of parameter values for each scenario) provides a significantly better fit than pooled analysis (e.g., having a shared set of parameter values common to all three scenarios). Alternatively, you could compare the AIC of the pooled data model with the AIC for the three models with unpooled data.

As explained in Line 265, the purpose of pooling the data (“Proglottids”, “Eggs” and “Beetles” pathways) to build the “Oral” pathway was not necessarily the obtainment of a model with a better fit but a robust model with less uncertainty across the range of doses assessed which represents the natural infection pathway of the disease.

Due to the increasement of observations in this model, the “Oral” pathway does not have better fit scores when comparing the Sum of Squared Errors of Prediction and the Coefficient of Determination calculated with the three individual models in most of the scenarios (development of any, viable and brain cysts) (see S1 Appendix, we have added the fit scores of the exact Beta-Poisson model for the “Oral” pathway). However, this model is more robust than the other three pathways and it is evidenced in the uncertainty presented in Figure 2 across the range of observations.

For these reasons, we are not considering evaluating whether the “Oral” model provides a better fit than the individual pathways (“Proglottids”, “Eggs” and “Beetles” pathways).

• Table 3 – It is typical to report maximum likelihood estimates of parameters rather than medians.

Thanks for letting us notice this point. We have replaced the medians by the MLEs of parameters for both the exact and the approximate beta-Poisson models.

• Line 387 – Appendix S2

Thank you for letting us notice the mistake. We have corrected the in-text citation.

• There is a very small number of grammatical issues throughout:

- Line 60: Revise to “…each of the four…”

- Line 77: Revise to “…models have been developed.” (to clarify that such models were not part of this work)

- Line 107: Revise to “surrounding”

- Lines 158-162: It is problematic to have two colons in this long sentence, and use “operators”.

- Lines 281, 283: Revise to “cyst”

Thank you for letting us notice these grammatical issues. We have corrected the phrases as suggested.

---

## [Editor Report · Decision Letter 2]

21 Feb 2022

Development of a dose-response model for porcine cysticercosis

PONE-D-21-23111R2

Dear Dr. Gonzales-Gustavson,

We’re pleased to inform you that your manuscript has been judged scientifically suitable for publication and will be formally accepted for publication once it meets all outstanding technical requirements.

Kind regards,

Brecht Devleesschauwer

Academic Editor

PLOS ONE
---

## [Editor Report · Acceptance letter]

28 Feb 2022

PONE-D-21-23111R2 

Development of a dose-response model for porcine cysticercosis 

Dear Dr. Gonzales-Gustavson:

I'm pleased to inform you that your manuscript has been deemed suitable for publication in PLOS ONE. Congratulations! Your manuscript is now with our production department. 

Kind regards, 

on behalf of

Prof. Dr. Brecht Devleesschauwer 

Academic Editor

PLOS ONE